# Screening identifies small molecules that enhance the maturation of human pluripotent stem cell-derived myotubes

Sridhar Selvaraj[1†], Ricardo Mondragon-Gonzalez[1,2†], Bin Xu[3], Alessandro Magli[1,4], Hyunkee Kim[1], Jeanne Lainé[5], James Kiley[1], Holly Mckee[1], Fabrizio Rinaldi[6], Joy Aho[6], Nacira Tabti[5], Wei Shen[1,3,4], Rita CR Perlingeiro[1,4]*

[1]Lillehei Heart Institute, Department of Medicine, University of Minnesota, Minneapolis, United States; [2]Departamento de Genética y Biología Molecular, Centro de Investigación y de Estudios Avanzados del IPN (CINVESTAV-IPN), Ciudad de México, Mexico; [3]Department of Biomedical Engineering, University of Minnesota, Minneapolis, United States; [4]Stem Cell Institute, University of Minnesota, Minneapolis, United States; [5]Département de Physiologie, Sorbonne Universités, Faculté de Médecine site Pitié-Salpêtrière, Paris, France; [6]Stem Cell Department, Bio-Techne, Minneapolis, United States

*For correspondence:
perli032@umn.edu

[†]These authors contributed equally to this work

**Abstract** Targeted differentiation of pluripotent stem (PS) cells into myotubes enables in vitro disease modeling of skeletal muscle diseases. Although various protocols achieve myogenic differentiation in vitro, resulting myotubes typically display an embryonic identity. This is a major hurdle for accurately recapitulating disease phenotypes in vitro, as disease commonly manifests at later stages of development. To address this problem, we identified four factors from a small molecule screen whose combinatorial treatment resulted in myotubes with enhanced maturation, as shown by the expression profile of myosin heavy chain isoforms, as well as the upregulation of genes related with muscle contractile function. These molecular changes were confirmed by global chromatin accessibility and transcriptome studies. Importantly, we also observed this maturation in three-dimensional muscle constructs, which displayed improved in vitro contractile force generation in response to electrical stimulus. Thus, we established a model for in vitro muscle maturation from PS cells.
DOI: https://doi.org/10.7554/eLife.47970.001

## Introduction

Pluripotent stem (PS) cells represent an attractive model system for disease modeling, drug screening and cell therapy applications for genetic diseases. PS cells possess the unique feature of unlimited proliferative potential and the ability to differentiate into all cell types of the body. The advent of induced pluripotent stem (iPS) cell technology enables the easy derivation of PS cells from any individual, including patients with genetic diseases (*Takahashi et al., 2007*). This allows for the generation of unlimited numbers of patient-specific cell derivatives, which can potentially recapitulate a given disease phenotype (*Avior et al., 2016*). For this to be successfully accomplished, two critical aspects need to be fulfilled. One is the use of an efficient methodology for generating the cell type of interest. The second critical aspect is maturation. This latter point in particular has been a conundrum in the field since extensive literature suggests that PS cell-derivatives are predominantly embryonic in nature (*Abdelalim and Emara, 2015*; *Aigha and Raynaud, 2016*; *Chen et al., 2018*; *Jiwlawat et al., 2018*).

To date, attempts to induce maturation of PS cell-derivatives involve maintaining the cells in culture for long periods of time, generally longer than one month (*Lainé et al., 2018*; *Paavilainen et al., 2018*; *Zhang et al., 2009*). A recent study comparing the transcriptional profile of transgene-free PS cell-derived myotubes with fetal myotubes revealed inhibition of TGFβ signaling as an approach to enhance in vitro PS cell-derived skeletal muscle maturation (*Hicks et al., 2018*). Notably, inhibition of the TGFβ signaling pathway through the use of small molecules has been extensively reported to result in enhanced myotube differentiation and hypertrophy (*Furutani et al., 2011*; *Liu et al., 2001*; *Schabort et al., 2009*; *Watt et al., 2010*). These studies suggest that small molecules have the potential to modify the profile of differentiation and maturation of skeletal muscle cells in vitro.

In this study, we performed a small molecule screening and identified four candidates whose combinatorial treatment enhances the maturation of PS cell-derived myotubes in a time frame of 5 days of myotube differentiation. This was evident by the striking increase in the expression profile of neonatal myosin heavy chain (MyHC) isoform as well as in other genes associated with neonatal and adult muscle, as revealed by chromatin accessibility and RNA sequencing analyses. Moreover, we demonstrate the usefulness of the combinatory small molecule treatment for inducing maturation and improving contractile force generation of three-dimensional (3D) PS cell-derived muscle constructs, thus contributing to the use of organoid-like platforms for in vitro modeling.

## Results

### Optimal differentiation of PS cells into somite-like stage enhances PAX7-induced myogenesis

To induce mesoderm differentiation of PS cells, we utilized CHIR99021 (GSK3β inhibitor) treatment for two days (*Figure 1—figure supplement 1A*), which we and others have previously reported to improve skeletal myogenic differentiation from PS cells (*Borchin et al., 2013*; *Kim et al., 2016*). This treatment resulted in the expression of the early mesoderm marker *T* as well as the paraxial mesoderm markers *MSGN1* and *TBX6* (*Figure 1—figure supplement 1B*). Considering the recent literature demonstrating the ability of the BMP inhibitor LDN193189 and the TGFβ inhibitor SB431542 to induce somitic mesoderm-like cells (*Xi et al., 2017*), we investigated whether these inhibitors would enhance myotube generation in the context of PAX7-induced myogenic differentiation. Treatment of differentiating PS cells from day 4 to day 6 with LDN193189 and SB431542 (+LS) (*Figure 1—figure supplement 1A*) resulted in increased expression of *MEOX1*, *TCF15*, *PAX3* and *FOXC2* on day 6 (*Figure 1—figure supplement 1B*). Induction of PAX7 expression with doxycycline began on day 5, two days earlier than our standard protocol (*Darabi et al., 2012*), as we reasoned that optimal myogenic specification by PAX7 would be achieved if it was induced when cells are at the peak of somite-like state. On day 12, PAX7$^+$ myogenic progenitors were purified based on GFP expression, expanded in the presence of doxycycline and bFGF for three cell passages, and then subjected to terminal differentiation culture conditions, as described previously (*Darabi et al., 2012*). Of note, MyHC-expressing myotubes were detected only when cultures were subjected to terminal differentiation following withdrawal of doxycycline. Our results showed significant improvement in the differentiation efficiency of several of the seven PS cell lines investigated (unaffected and diseased), when compared to the unmodified protocol (-LS) (*Figure 1—figure supplement 2*). This result was particularly evident in PS cell lines displaying limited in vitro differentiation potential using the unmodified protocol.

### Small molecule library screening for enhancing myogenic differentiation/maturation

Despite the promising results described above, PS cell-derived myotubes remained immature, as indicated by their thin morphology (*Figure 1—figure supplement 2*) and predominant expression of the embryonic isoform of myosin heavy chain (*MYH3*) (*Figure 1A*). To determine whether small molecule compounds may enhance the maturation of PS cell-derived myotubes, we performed a small molecule library screening using the Tocriscreen stem cell toolbox kit (Tocris). This library consists of 80 stem cell modulator compounds. Myogenic progenitors were seeded onto 96-well plates and incubated for three days in the presence of expansion medium (bFGF and Dox). Then, culture

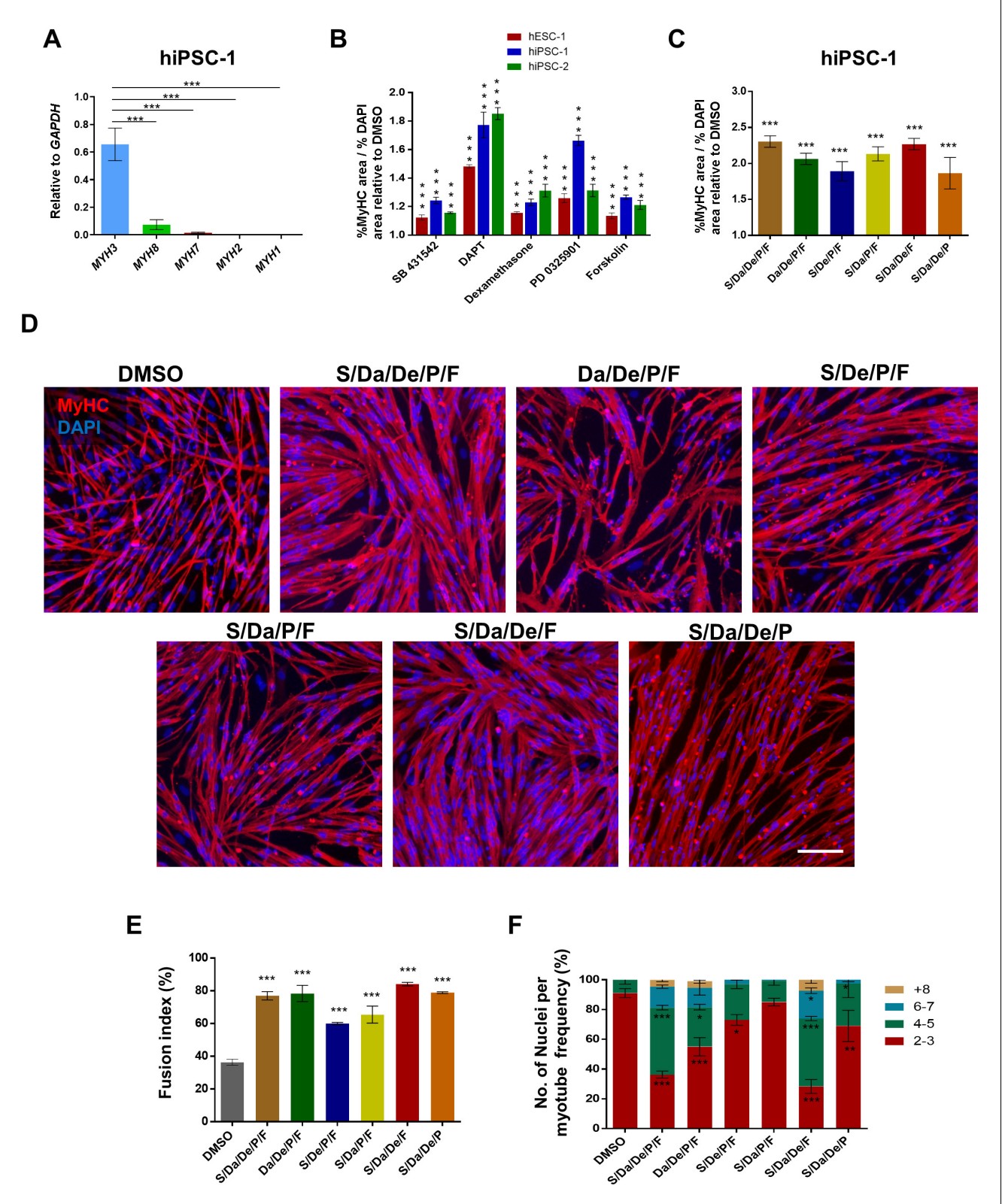

**Figure 1.** Combinatorial treatment with four small molecules augments myotube generation from human PS cells and their fusion ability. (A) Bar graph shows expression profile of *MYH* isoforms in hiPSC-1-derived myotubes. Data are shown as mean ± S.E.M.; n = 3, ***p<0.001. (B) Bar graph shows the ratio of % MyHC-stained area to % DAPI area in myotubes resulting from treatment with five candidates identified by the small molecule screening. Data show significant increase (***p<0.001) compared to DMSO in all three PS cell lines analyzed (hESC-1, hiPSC-1 and hiPSC-2). Data from three

*Figure 1 continued on next page*

*Figure 1 continued*

independent replicates are shown, normalized to DMSO, as mean ± S.E.M. (**C**) Bar graph shows the ratio of % MyHC-stained area to % DAPI area in iPS cell-derived myotubes that had been differentiated in the presence of all candidates combined, or with individual candidates excluded from the overall combination. Data from three independent replicates are shown normalized to DMSO. Values are shown as mean ± S.E.M. ***p<0.001. (**D**) Representative images show immunostaining for MyHC (in red) in hiPSC-1 myotubes differentiated with combinatory treatments of small molecules or DMSO. DAPI stains nuclei (in blue). Scale bar is 100 μm. (**E**) Bar graph shows fusion index analysis of myotubes that were differentiated with small molecule combinations or DMSO. Data are shown as mean of three independent replicates ± S.E.M. ***p<0.001. (**F**) Stacked bar graph shows the frequency of number of nuclei per myotube upon differentiation with combinatory treatments or DMSO. Data are shown as mean of three independent replicates ± S.E.M. Statistical analysis compares each combination to DMSO. *p<0.05 **p<0.01 ***p<0.001.
DOI: https://doi.org/10.7554/eLife.47970.002

The following source data and figure supplements are available for figure 1:

**Source data 1.** Tocriscreen Stem Cell Toolbox compounds tested during myogenic terminal differentiation of PS cell lines.
DOI: https://doi.org/10.7554/eLife.47970.006
**Figure supplement 1.** BMP and TGFβ signaling inhibition induce somite-like specification during the in vitro muscle differentiation of iPAX7 PS cells.
DOI: https://doi.org/10.7554/eLife.47970.003
**Figure supplement 2.** Induction of somite-like stage enhances iPAX7 PS cell-derived myogenic differentiation into myotubes.
DOI: https://doi.org/10.7554/eLife.47970.004
**Figure supplement 3.** Small molecule screening reveals compounds that enhance myogenic differentiation efficiency.
DOI: https://doi.org/10.7554/eLife.47970.005

medium was switched to differentiation medium supplemented or not with small molecules from the library, with each well containing an individual compound in a final concentration of 10 μM. Five days later, cells were stained for MyHC expression (*Figure 1—figure supplement 3A*). We reasoned that an increase in the ratio of MyHC (+) to DAPI (+) area would indicate thicker myotubes as a consequence of increased multinucleation, hypertrophy and/or enhanced differentiation, which could be potentially accompanied by increased maturation (*Biressi et al., 2007b*). We identified several compounds exhibiting a positive or negative effect on terminal differentiation (*Figure 1—source data 1*). For the purposes of this study, we selected five compounds that consistently showed statistically significant increase (p<0.001) in MyHC/DAPI area ratio in myotubes derived from multiple PS cell lines (*Figure 1B*). This set includes the TGFβ signaling inhibitor SB431542 (S), the γ-Secretase inhibitor DAPT (Da), the anti-inflammatory glucocorticoid Dexamethasone (De), the MEK inhibitor PD0325901 (P), and the adenylyl cyclase activator Forskolin (F) (*Figure 1B* and *Figure 1—figure supplement 3B*). Since concentrations ranging from 5 to 20 μM of the selected compounds did not significantly change the MyHC/DAPI area ratio (*Figure 1—figure supplement 3C*), in subsequent studies we used 10 μM for each compound.

Expression of *MYH* isoforms can be used as a readout for the maturation state of generated myotubes, as each *MYH* isoform is found predominantly expressed at specific developmental stages: *MYH3* for embryonic, *MYH8* for neonatal, and *MYH1*, *MYH2* and *MYH7* for adult muscle (*Schiaffino et al., 2015*). Therefore, we analyzed the *MYH* expression profile for each of the individual compound treatments. Although there was an overall increase in the expression of embryonic, neonatal and adult *MYH* isoforms among compounds, we did not detect a trend of maturation shift for a particular candidate (*Figure 1—figure supplement 3D*). Therefore, we assessed the combination of all five compounds (S/Da/De/P/F) along with combinations obtained by excluding each compound from the overall mixture. Significant increase in MyHC/DAPI area ratio (>2 fold relative to DMSO) was observed in all combinations (*Figure 1C*). Morphologically, myotubes exposed to S/Da/De/P/F treatment appeared thicker and denser relative to DMSO. Exclusion of S, Da, De or F led to the generation of thinner myotubes compared with the full combinatorial condition, while exclusion of P did not affect myotube morphology (*Figure 1D*). Consistently, this correlated with remarkable increases in fusion index and number of nuclei per myotube in the S/Da/De/P/F condition, which again were unaffected when excluding P but not the other compounds (*Figure 1E–F*).

## Combinatorial treatment induces maturation switch

To determine whether enhanced myogenic differentiation, multinucleation and changes in myotube morphology are accompanied by molecular changes, we performed gene expression analysis for Myogenin (*MYOG*) and several *MYH* isoforms. Although S/Da/De/P/F or S/Da/De/P combinatorial

conditions showed significant increase for some *MYH* isoforms, our data demonstrated that only S/Da/De/F (excluding P) consistently resulted in significantly increased expression of all *MYH* isoforms analyzed, with a more dramatic upregulation observed for neonatal *MYH8* (neo-MyHC) (*Figure 2A*). Based on these results, we selected the S/Da/De/F condition for subsequent studies. We tested S/Da/De/F combinatorial treatment on additional differentiating PS cell lines, and consistently observed increased expression of *MYH8* (*Figure 2B*). Of note, significant increase in the gene expression of adult isoforms *MYH2* and *MYH7* was observed in myotubes derived from iPSC-2, while ES cell-derived myotubes remained unchanged for these isoforms (*Figure 2B*). Western blot and immunostaining analysis confirmed expression of neo-MyHC at the protein level only upon S/Da/De/F treatment (*Figure 2C–D*), whereas MYH1 and MYH2 were not detected upon treatment (*Figure 2C*). Furthermore, immunostaining revealed cross-striation staining pattern of neo-MyHC co-localizing with that of F-actin (stained with Phalloidin), suggesting protein functionality (*Figure 2— figure supplement 1A*). Importantly, neo-MyHC protein expression was confirmed in additional three PS cell lines subjected to the combinatorial treatment (*Figure 2—figure supplement 1B*). Of relevance, a previous study reported that neo-MyHC protein expression only occurs at later stages of human muscle development, thus further confirming the enhanced maturation status of myotubes treated with S/Da/De/F (*Cho, 1993*). As shown by Titin immunofluorescence staining (*Figure 2E*), both DMSO- and S/Da/De/F-treated myotubes are striated. However, S/Da/De/F treatment resulted in thicker and larger myotubes (*Figure 2E*).

To determine the temporal window by which small molecule treatment enhances in vitro terminal differentiation, we analyzed proliferation and differentiation parameters at days 1, 3, and 5 of terminal differentiation by EdU staining and MyHC/DAPI ratio, respectively. Interestingly, whereas no differences were observed between DMSO and S/Da/De/F at days 1 and 3 for both parameters, at day 5, we observed dramatic reduction in cell proliferation and increase in differentiation in the S/Da/De/F group (*Figure 2—figure supplement 1C–D*). These data suggest that the small molecule treatment may be particularly relevant at the late stage of differentiation. Next, we analyzed the expression of genes related to the pathways targeted by each of the small molecules (S, Da, De and F) at day 5 of myotube differentiation. As shown in *Figure 2—figure supplement 2A–D*, we found significant differential expression of *CEBPD* and *FKBP5* (De targets; *Luedi et al., 2017*; *MacDougald et al., 1994*; *Pan et al., 2010*; *Pereira et al., 2014*; *Reddy et al., 2012*; *Scharf et al., 2011*), *HES1* and *NOTCH2* (Da targets; *Huang et al., 2011*; *Jarriault et al., 1995*; *Mao et al., 2018*), *COL1A1* and *ID3* (S targets; *Lehmann et al., 2018*; *Ramachandran et al., 2018*; *Sato et al., 2015*) and *PPARGC1A* (F target; *Charos et al., 2012*; *Martin et al., 2009*; *Sayasith et al., 2014*). Of note, S/Da/De/F treatment also induced the maturation of PS cell-derived myotubes generated under transgene-free conditions (*Xi et al., 2017*) (*Figure 2—figure supplement 3A–B*).

Based on the relevance of these findings for the disease modeling of muscular dystrophies (MD), we then investigated the effect of this small molecule combination on the differentiation and maturation of myotubes derived from a panel of MD patient-specific iPS cells, including two Duchenne Muscular Dystrophy (DMD1 and DMD2), two Myotonic Dystrophy type 1 (DM1-1 and DM1-2) and one LGMD2A (Key resources table) (*Magli et al., 2017*; *Mondragon-Gonzalez and Perlingeiro, 2018*; *Selvaraj et al., 2019*). Consistently, our data showed enhanced differentiation (*Figure 3A*) and maturation (*Figure 3B*) of MD patient-specific iPS cell-derived myotubes upon S/Da/De/F treatment. These findings confirm the positive effects of this combinatorial treatment in the differentiation and maturation of human iPS cell-derived myotubes.

## Ultrastructural differences between DMSO and S/Da/De/F-treated PS-derived skeletal myotubes

Transmission electron microscopy was performed on ultrathin sections of PS cell-derived myotubes differentiated for 19 days in the absence or presence of small molecules. Analysis of a large number of samples at high magnification showed that myofibrils reached various degrees of sarcomeric organization even within the same cell in both conditions (*Figure 4*). This may be explained by ongoing fusion between cells with different maturation levels or stages (see below). Examples of well-differentiated sarcomeres with clearly delineated Z-band, alternating A-I bands, and M-line crossing the H-zone are shown in *Figure 4B and F*. Intermediate stages of sarcomeric formation with discernable A-I banding pattern, but discontinuous Z-band are depicted in *Figure 4C and G*, and nascent sarcomeres with undefined banding pattern and Z-bodies are shown in *Figure 4D and H*. S/Da/De/F-

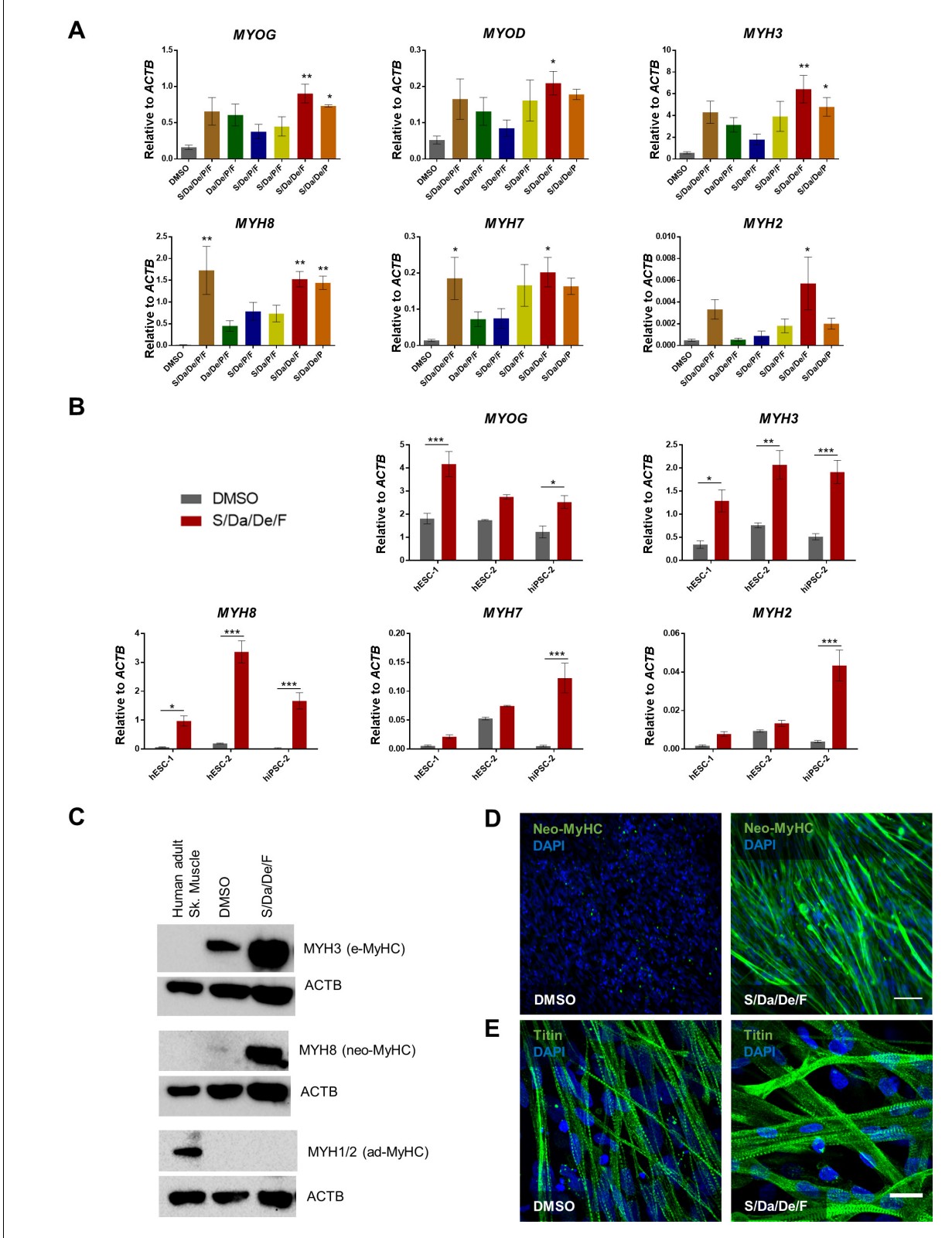

**Figure 2.** Combinatorial treatment with S/Da/De/F enhances the maturation of PS cell-derived myotubes. (**A**) Bar graphs show the expression profile of *MYOG, MYOD* and *MYH* isoforms normalized to *ACTB* in hiPSC-1 myotubes differentiated with small molecule combinatorial treatment or DMSO. Data are shown as mean of three independent replicates ± S.E.M. *p<0.05 **p<0.01. (**B**) Bar graphs show expression levels of *MYOG,* and *MYH* isoforms normalized to *ACTB* in hESC-1, hESC-2 and hiPSC-1 myotubes differentiated with combinatory treatment or DMSO. Data are shown as mean of three
*Figure 2 continued on next page*

*Figure 2 continued*

independent replicates ± S.E.M. *p<0.05 **p<0.01. (C) Western blot shows protein expression for MYH3 (e-MyHC), MYH8 (neo-MyHC) and MYH1/2 (ad-MyHC) in hiPSC1 myotubes that had been subjected to treatment with S/Da/De/F or DMSO. Human adult skeletal muscle is shown as a reference. ACTB is used as loading control. (D, E) Representative images show immunostaining for neo-MyHC (in green) (D) and Titin (in green) (E) in hiPSC-1 myotubes differentiated in the presence of DMSO or S/Da/De/F. DAPI stains nuclei (blue). Scale bars are 100 µm (D) and 20 µm (E).

DOI: https://doi.org/10.7554/eLife.47970.007

The following figure supplements are available for figure 2:

**Figure supplement 1.** Combinatorial treatment promotes neo-MyHC protein expression.

DOI: https://doi.org/10.7554/eLife.47970.008

**Figure supplement 2.** Combinatorial treatment targets pathways associated with its individual components.

DOI: https://doi.org/10.7554/eLife.47970.009

**Figure supplement 3.** Combinatorial small molecule treatment enhances the maturation of hiPS cell-derived myotubes generated under transgene-free differentiation conditions.

DOI: https://doi.org/10.7554/eLife.47970.010

treated myotubes were clearly richer in mitochondria than their DMSO counterparts. As illustrated in *Figure 4E and I*, mitochondria were often densely packed between the myofibrils, or spread along the plasma membrane. *Figure 4I–L* focuses on the high occurrence of SR-TT junctions following the treatment with small molecules. Indeed, these structures were readily detectable in all S/Da/De/F-treated myotubes (*Figure 4* and *Figure 4—figure supplement 1D*), while they remained scarce in the DMSO-treated myotubes (*Figure 4D*). Internal junctions were usually multiple and varied in shape, size and location. They were not associated with the myofibrils and had rarely a triadic configuration. Similar junctions were shown to produce voltage-dependent intracellular $Ca^{2+}$ transients upon membrane depolarization, and hence, support excitation-contraction coupling (*Skoglund et al., 2014*). Therefore, the high occurrence of SR-TT junctions in treated cells may contribute to the increase in their contraction capacity.

Ultrastructural examination of adjacent cells at high magnification revealed ongoing plasma membrane fusion in both conditions. Such a process was, however, undoubtedly more prolific in S/Da/De/F-treated cells (*Figure 4—figure supplement 1*). Indeed, a larger number of S/Da/De/F-treated cells were engaged in fusion (>5 cells) as compared with mainly 2 (or rarely 3) in the control. Fusion paths between 2 DMSO treated and 5 S/Da/De/F-treated cells are highlighted in *Figure 4—figure supplement 1B and D*, respectively. This goes in line with the increase in fusion index and number of myonuclei observed in the S/Da/De/F group (*Figure 1E–F*).

## Increased chromatin accessibility at myogenic loci upon small molecule treatment

To dissect the mechanism behind S/Da/De/F-mediated myogenic maturation, we investigated the chromatin accessibility landscape using ATAC-seq (*Buenrostro et al., 2013*). Following a 2 day treatment, we detected 62,748 and 80,334 chromatin accessible peaks in DMSO- and S/Da/De/F-treated cells, respectively (*Figure 5A*). Comparison of these two datasets showed an overlap of 20,682 peaks, while 42,066 and 59,652 peaks were specific for DMSO and S/Da/De/F treatments, respectively. To determine whether S/Da/De/F treatment led to significant changes in chromatin accessibility, we next compared the normalized sequencing depth at chromatin accessible peaks identified in all replicates. Principal component analysis (PCA) showed that DMSO- and S/Da/De/F-treated groups clustered separately (*Figure 5—figure supplement 1A*). Differential analysis of chromatin accessibility followed by exclusion of loci overlapping to centromeric and other repetitive elements (referred as blacklist; *ENCODE Project Consortium, 2012*) identified 2782 peaks displaying increased chromatin accessibility in the S/Da/De/F-treated group (*Figure 5B*). As expected, this group included genomic elements associated with myogenic genes such as *MYOG*, *MYH3* and the muscle microRNAs *MIR206* and *MIR133B* (*Chen et al., 2006*; *Kim et al., 2006*; *Koutsoulidou et al., 2011*; *Swiderski et al., 2016*) (*Figure 5C* and *Figure 5—figure supplement 1B*). Similarly, increased chromatin accessibility was detected at the *CEBPD* locus (*Figure 5C*), a Dexamethasone target gene in human cells (*MacDougald et al., 1994*; *Pan et al., 2010*; *Reddy et al., 2012*). In agreement with the biological function of SB431542 and DAPT, we also observed significant reduction in chromatin accessibility at loci encoding or regulating downstream target genes of the TGFβ (*ID3*

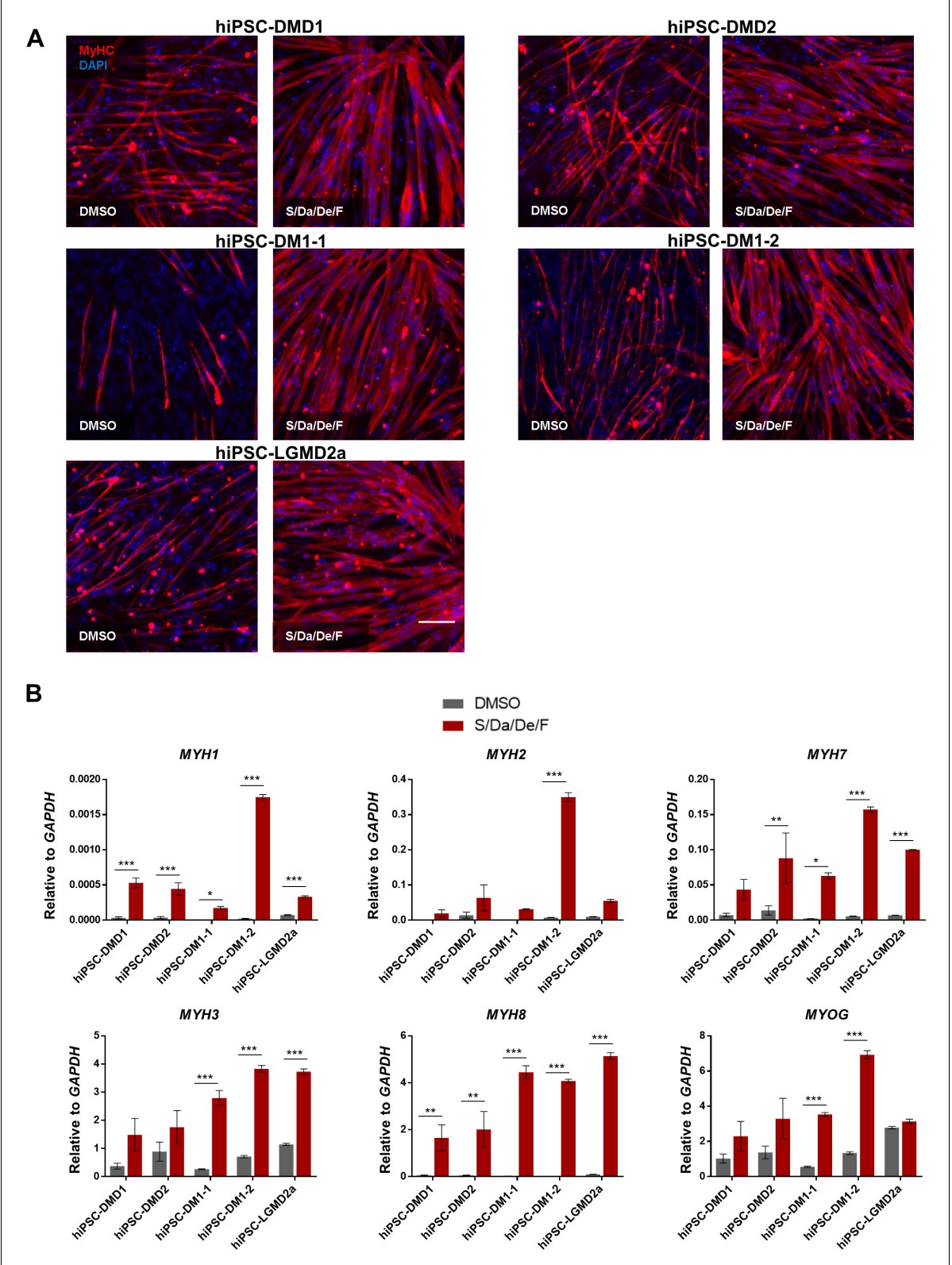

**Figure 3.** Combinatorial treatment with S/Da/De/F enhances the maturation of MD patient-specific hiPS cell-derived myotubes. (A) Representative images show immunostaining for MyHC (in red) in hiPS cell-derived myotubes from two DMD (DMD1 and DMD2), two DM1 (DM1-1 and DM1-2) and one LGMD2A patients differentiated with small molecule combinatorial treatment or DMSO. DAPI stains nuclei (blue). Scale bar is 100 μm. (B) Bar graphs show the expression profile of *MYH* isoforms and *MYOG* isoforms normalized to *GAPDH* in hiPS cell-derived myotubes from two DMD (DMD

*Figure 3 continued on next page*

*Figure 3 continued*

one and DMD 2), two DM1 (DM1-1 and DM1-2) and one LGMD2A patients differentiated with small molecule combinatorial or DMSO treatments. Data are shown as mean of three independent replicates ± S.E.M. *p<0.05 **p<0.01 ***p<0.001.

DOI: https://doi.org/10.7554/eLife.47970.011

The following figure supplement is available for figure 3:

**Figure supplement 1.** Characterization of hiPSC-3, hiPSC-4 and hiPSC-DMD1 reprogrammed cell lines.

DOI: https://doi.org/10.7554/eLife.47970.012

(*Ramachandran et al., 2018*) and NOTCH (*HES1*) (*Jarriault et al., 1995*) signaling pathways. (*Figure 5—figure supplement 1B*).

To further characterize these elements, we analyzed the S/Da/De/F specific regions for the presence of conserved DNA binding motifs. This analysis showed enrichment for several motifs associated with known myogenic transcription factors, including MYOG, MEF2, and POU (*Figure 5D–E*). Among these, the POU transcription factor Pou2f1 (also known as Oct1), through cooperation with SRF and MEF2, regulates transcription of the fast MyHC (MyHC-2b) (*Allen et al., 2005*). Analogously, Pou6f1 (also called Emb) was identified in a complex with MEF2D and p300, whose function is to control Actc1, an actin isoform expressed in developing skeletal muscle and cardiac muscle (*Molinari et al., 2004*).

## Transcriptomic analysis reveals genes associated with muscle differentiation, maturation and contractile function upon small molecules treatment

To further define the molecular changes induced by S/Da/De/F, we performed RNA sequencing analysis on differentiated myotubes obtained upon treatment with small molecules and DMSO control. We identified about 1859 significantly differentially expressed genes, from which about half were downregulated and the other half were upregulated in the treated group compared to controls (*Figure 6A*). To dissect the transcriptome profile associated with S/Da/De/F treatment, we first performed ingenuity pathway analysis (IPA) and corroborated the four small molecules (and their targets) as upstream regulators of the differentially expressed genes, which was evidenced by the 'activation Z-scores' (<-2 or >2) and validated by the overlap p-values (<0.01) (*Figure 6—source data 1*). Interestingly, the analysis also revealed transcription factors and miRNAs involved in muscle differentiation as upstream regulators of changes in gene expression upon S/Da/De/F treatment, including *TEAD4* (*Joshi et al., 2017*), *MYF6* (*Kassar-Duchossoy et al., 2004*; *Montarras et al., 1991*), *KLF4* (*Sunadome et al., 2011*), *MEF2A*, *MEF2C* (*Liu et al., 2014*), *MIR-133B*, *MIR-206* and *MIR-503* (*Chen et al., 2006*; *Kim et al., 2006*; *Koutsoulidou et al., 2011*; *Sarkar et al., 2010*) (*Figure 6B*). The differentially expressed targets of these transcription factors and miRNAs, are listed in *Figure 6—source data 2*. According to our IPA data, skeletal and muscular system development was found the most upregulated physiological system (*Figure 6C*). In addition, among upregulated canonical pathways, the analysis identified oxidative phosphorylation and calcium signaling, indicating an increase in mitochondrial biogenesis and muscle contraction, respectively (*Figure 6D*), corroborating the increased number of mitochondria observed by TEM (*Figure 4*). This makes sense as skeletal myogenesis has been documented to be accompanied by mitochondrial biogenesis (*Remels et al., 2010*; *Sin et al., 2016*). Moreover, Sarcolipin (*SLN*), a gene reported to play a critical role in mitochondrial biogenesis and oxidative metabolism in skeletal muscle (*Maurya et al., 2018*), was found among the highest expressed genes (around 30-fold) in the treated group (a result we have validated by qPCR; *Figure 6—figure supplement 1A*). Gene ontology analysis of upregulated genes for biological processes and cellular components identified various muscle function groups, indicating increased muscle differentiation and function (*Figure 6E–F*). Consistently, we found upregulation of genes, such as *ENO3*, *MYF6*, *CKM*, *TNNT3*, *MYH8*, *ATP2A1*, *ITGA7* and *PRKCQ*, all reported to be upregulated in fetal, postnatal and adult myotubes in comparison to embryonic myotubes (*Biressi et al., 2007a*; *Burkin and Kaufman, 1999*; *Hinterberger et al., 1991*; *Ju et al., 2013*; *Schiaffino et al., 2015*), supporting our observations on the enhancing effect of the combinatorial treatment on myotubes maturation. We validated the upregulation of some of these genes by RT-qPCR (*Figure 6—figure supplement 1A*). Correspondingly, we found that S/Da/De/F-treated

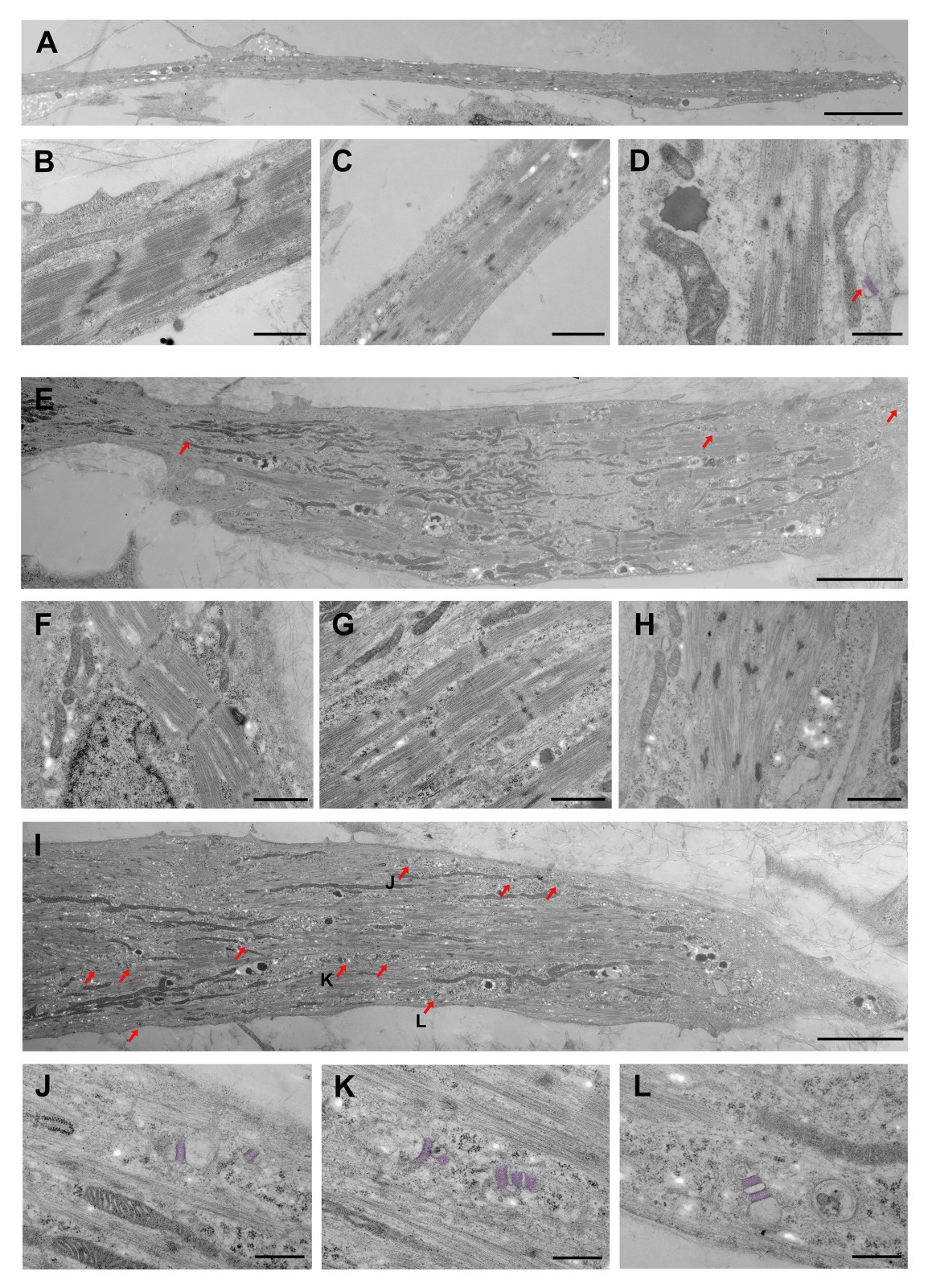

**Figure 4.** Ultrastructural differences between S/Da/De/F- and DMSO-treated myotubes. (**A–D**) DMSO-treated myotubes (control) are shown at different magnifications. (**A**) Low magnification shows a thin and elongated myotube. (**B, C**) Myofibrils from control myotubes display different degrees of sarcomeric organization. (**B**) Discernable A-I bands, M line, and winding Z-bands, (**C**) Incomplete banding pattern and Z-bodies. (**D**) High magnification shows one SR-TT junction at the periphery of the cell. The SR has been highlighted by artificial post-coloring. (**E–H**) S/Da/De/F treated myotubes are

*Figure 4 continued on next page*

*Figure 4 continued*

shown at different magnifications. (**E**) Large myotube with relatively well-organized myofibrils located at the periphery or in close proximity to large bundles of mitochondria; red arrows indicate SR-TT junctions. (**F–H**) Myotubes subjected to S/Da/De/F treatment also display myofibrils with various degrees of sarcomeric organization. (**F**) Well defined A-I pattern, well delineated Z bands and visible M lines. (**G**) Alternating A and I bands, but discontinuous Z-band. (**H**) Nascent sarcomere with undefined banding patterned and Z–bodies. (**I**) Large myotube with a great number of SR-TT junctions (red arrows). Notice the presence of numerous mitochondria. The junctions identified by J, K and L letters are enlarged below (**J–L**) Different examples of representative SR-TT junctions; the SR is highlighted by artificial post-coloring. Scale bars: 5 µm in A, E and I; 1 µm in B, C and F-H; 500 nm in D and J-L.

DOI: https://doi.org/10.7554/eLife.47970.013

The following figure supplement is available for figure 4:

**Figure supplement 1.** Transmission electron microscopy reveals an enhanced fusion process and numerous SR-TT junctions in S/Da/De/F-treated PS cell-derived myotubes.

DOI: https://doi.org/10.7554/eLife.47970.014

myotubes showed downregulation of *MEOX1, PAX3, CDH11, EYA2* and *FST*, genes known to be expressed at higher levels in embryonic myotubes than in fetal myotubes (*Biressi et al., 2007b*) (*Figure 6—figure supplement 1B*).

Taking advantage of our epigenetic analysis, we next assessed whether changes in chromatin accessibility were associated with differential gene expression. For this analysis, we focused on the loci characterized by increased chromatin accessibility upon S/Da/De/F treatment. Upon annotation of the ATAC-seq peaks to the two nearest genes followed by comparison with the transcriptomic data, we observed that 42% of the loci were associated to differentially expressed genes (1727/4145 annotated genes). Of these, 878 and 849 transcripts were upregulated and downregulated, respectively (*Figure 6—figure supplement 2A*). Gene ontology classification of these two lists of genes confirmed that loci characterized by increased chromatin accessibility and increased expression are associated with muscle contraction, regulation of calcium storage and muscle development (*Figure 6—figure supplement 2B*). In contrast, loci with increased chromatin accessibility and decreased expression demonstrated enrichment for non-myogenic developmental pathways (*Figure 6—figure supplement 2C*). Based on these data, we conclude that S/Da/De/F treatment alters the epigenetic landscape of differentiating PS cell-derived myogenic progenitors by increasing chromatin accessibility at elements associated with key myogenic genes and pathways, potentially underlying the mechanism of small molecule-induced maturation of myotubes.

## Functional maturation using 3D muscle constructs

To determine the effect of the combinatorial treatment on the contractile function of resulting myotubes, we tested them on 3D muscle constructs that allow force measurement of PS cell-derived myotubes. Myogenic progenitors were seeded in 3D hydrogels containing fibrin and Matrigel to allow for expansion. After 3 days, the medium was switched to differentiation medium supplemented with DMSO or combinatorial treatment and cultured for 5 days (*Figure 7—figure supplement 1A*). At this point, the contractile force generation (twitch and tetanus) in response to electrical stimulus was measured. We found a remarkable increase in both twitch and tetanus force generation in the treated group (about 15-fold; *Figure 7A–B*). Gene expression analysis showed that treated-3D constructs expressed higher levels of various *MYH* isoforms (*Figure 7C*). Whereas we could not reliably detect upregulation of *MYH1* and *MYH2* gene expression levels in 2D cultures, we observed significant increase in expression of both genes upon treatment in 3D cultures (*Figure 7—figure supplement 1B*). Neo-MyHC was also upregulated at the protein level as revealed by western blot and immunostaining of the 3D muscle constructs (*Figure 7D* and *Figure 7—figure supplement 1C*), confirming superior differentiation and maturation. Increased pan-MyHC and α-actinin were observed as well by immunostaining of 3D constructs upon treatment (*Figure 7—figure supplement 1C*). In summary, the combinatorial treatment on 3D muscle constructs composed of PS cell-derived myotubes leads to higher levels of maturation and contractile force generation in response to electrical stimulus.

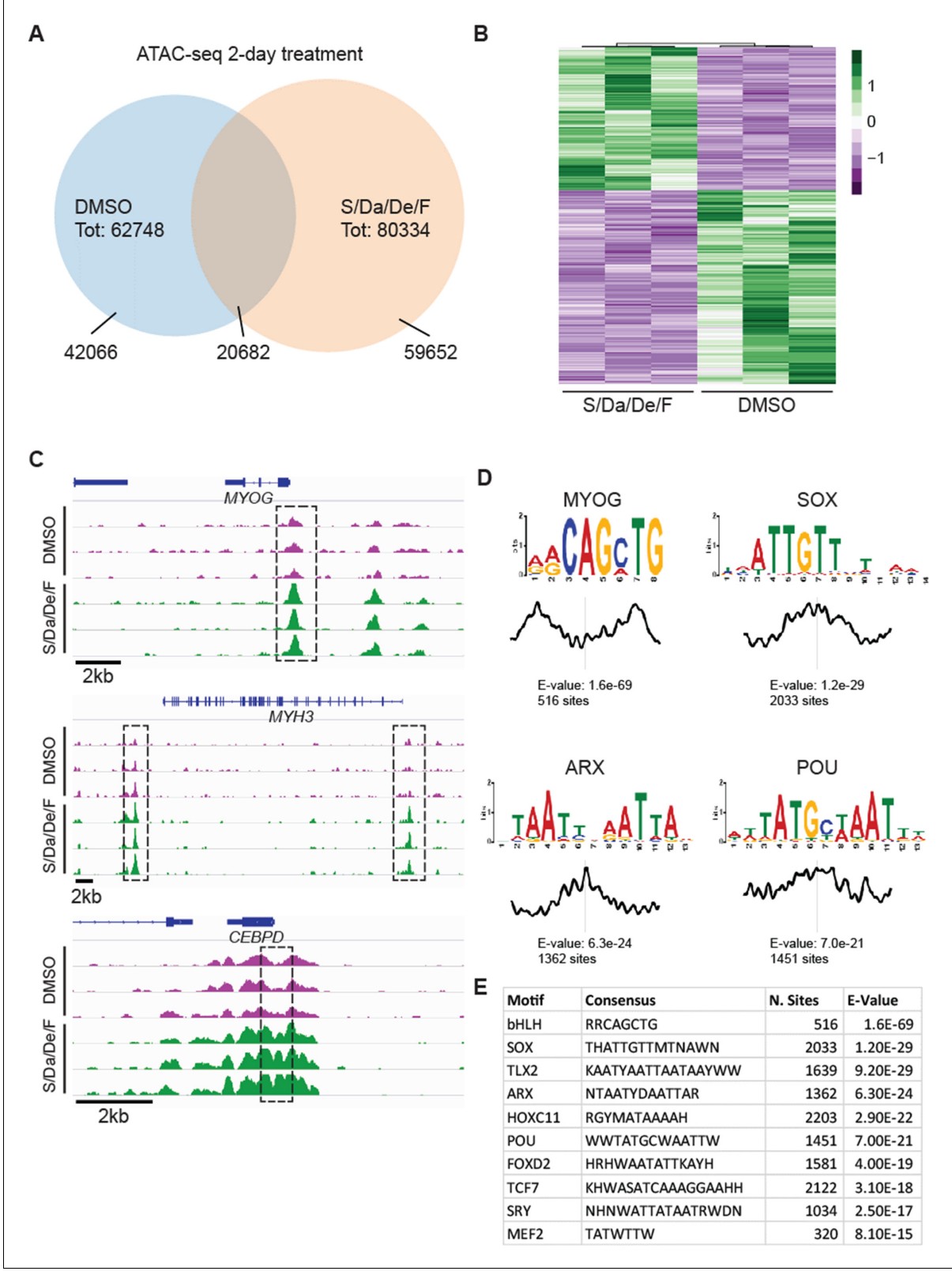

**Figure 5.** Combinatorial treatment increases chromatin accessibility at myogenic loci. (**A**) Venn diagram displaying overlap between loci detected in 2-day S/Da/De/F- and DMSO-treated cells. (**B**) Heatmap shows changes in chromatin accessibility between DMSO- and S/Da/De/F-treated cells (three independent biological replicates). Loci were selected based on adjusted p-value<0.05 and log2FoldChange > 1. Loci overlapping to blacklist regions are included in this heatmap. (**C**) Chromatin accessibility at the genomic loci proximal to *MYOG, MYH3 and CEBPD* genes. Dashed black boxes

*Figure 5 continued on next page*

*Figure 5 continued*

indicate loci characterized by significant change in chromatin accessibility. Tracks represent snapshots from the IGV browser. (**D**) Selected enriched motifs identified at S/Da/De/F-specific peaks using MEME-ChIP. Plot below the sequence logo indicates distribution of the motifs across the regions used as input. (**E**) Table schematizing the results obtained by MEME-ChIP. Only selected motifs are displayed.

DOI: https://doi.org/10.7554/eLife.47970.015

The following figure supplement is available for figure 5:

**Figure supplement 1.** Analysis of chromatin accessible peaks upon combinatorial treatment.

DOI: https://doi.org/10.7554/eLife.47970.016

## Discussion

The success of using PS cell derivatives for disease modeling and drug screening studies relies on producing a population of cells able to recapitulate the relevant structural and physiological features of the tissue under study. Skeletal muscle reaches full functionality through tissue maturation from embryonic, to fetal, neonatal and adult stages. These stages of maturation are characterized by the expression of specific proteins and protein isoforms, such as MyHC variants (*Schiaffino et al., 2015*). This is particularly relevant when studying muscular dystrophies with adult onset or where progression of the disease occurs at the post-neonatal stage. For instance, in Duchenne Muscular Dystrophy (DMD), absence of dystrophin at the sarcolemma is compensated to some extent by utrophin during early gestational stages (*Helliwell et al., 1992*; *Pons et al., 1993*). However, upon birth utrophin is located preferentially at the neuromuscular junction (*Ohlendieck et al., 1991*). Although in regenerating fibers utrophin can be detected at the sarcolemma (*Gramolini et al., 1999*), it is insufficient to sustain its stability, leading to the severe DMD phenotype in the adult muscle.

While several approaches have been reported to differentiate PS cells towards the myogenic lineage, there is still a limitation in promoting the full maturation of resulting myotubes. This issue compromises the biological relevance of in vitro studies data using PS cell- derivatives. Our small molecule screening in search for potential candidates able to promote maturation of PS cell-derived myotubes identified the compounds SB431542, DAPT, Dexamethasone, Forskolin and PD0325901 as potent enhancers of maturation, in particular when used in combination. Interestingly, SB431542 (TGFβ signaling inhibitor) and DAPT (Notch signaling inhibitor) have been described to promote hypertrophy and fusion of myoblasts lines, respectively (*Kitzmann et al., 2006*; *Watt et al., 2010*). Dexamethasone was an unexpected finding based on its well-known function in promoting atrophy in myotubes (*Hong and Forsberg, 1995*; *Thompson et al., 1999*; *Wang et al., 1998*). This suggests that dexamethasone may have an opposite role during differentiation of embryonic myotubes compared to adult counterparts, although a detailed mechanism remains to be further dissected. Forskolin has been reported to increase myoblast proliferation through activation of adenylate cyclase and subsequent phosphorylation and activation of cAMP response element binding protein (CREB) (*Stewart et al., 2011*). In contrast, our studies show that Forskolin treatment enhances the differentiation efficiency of PS cell-derived progenitors into myotubes. Finally, mitogen-activated protein kinase 1 (MEK1) has been shown to have contrary effects on myoblast differentiation depending on the time point of activation. For instance, MEK1 activation during the first 24 hr after switching to terminal differentiation conditions decreases differentiation while at a mid-stage stabilizes MyoD and promotes differentiation (*Jo et al., 2011*; *Jo et al., 2009*). We observed that MEK1 inhibition, through PD0325901, enhanced differentiation of PS cell-derived progenitors. Our findings corroborate the idea that although there are common factors involved in embryonic and adult myogenesis, the molecular signature of these processes is not identical (*Czerwinska et al., 2016*; *Lepper et al., 2009*; *Wang and Conboy, 2010*), and therefore, compounds should not be expected to cause the same effects.

Interestingly, we found that S/Da/De/F combination dramatically increased fusion, differentiation and maturation of resulting myotubes compared to individual treatments, and this coincided with increased chromatin accessibility in elements associated with myogenic specification at day 2, which suggests that treatment is beneficial from the beginning upon switching to terminal differentiation conditions. Moreover, RNA-seq analysis at day 5 revealed an increase in the expression of genes associated with muscle functionality and maturation processes. This is relevant as it shows that the combinatorial treatment is also beneficial to sustain myotube maturation. Importantly, the

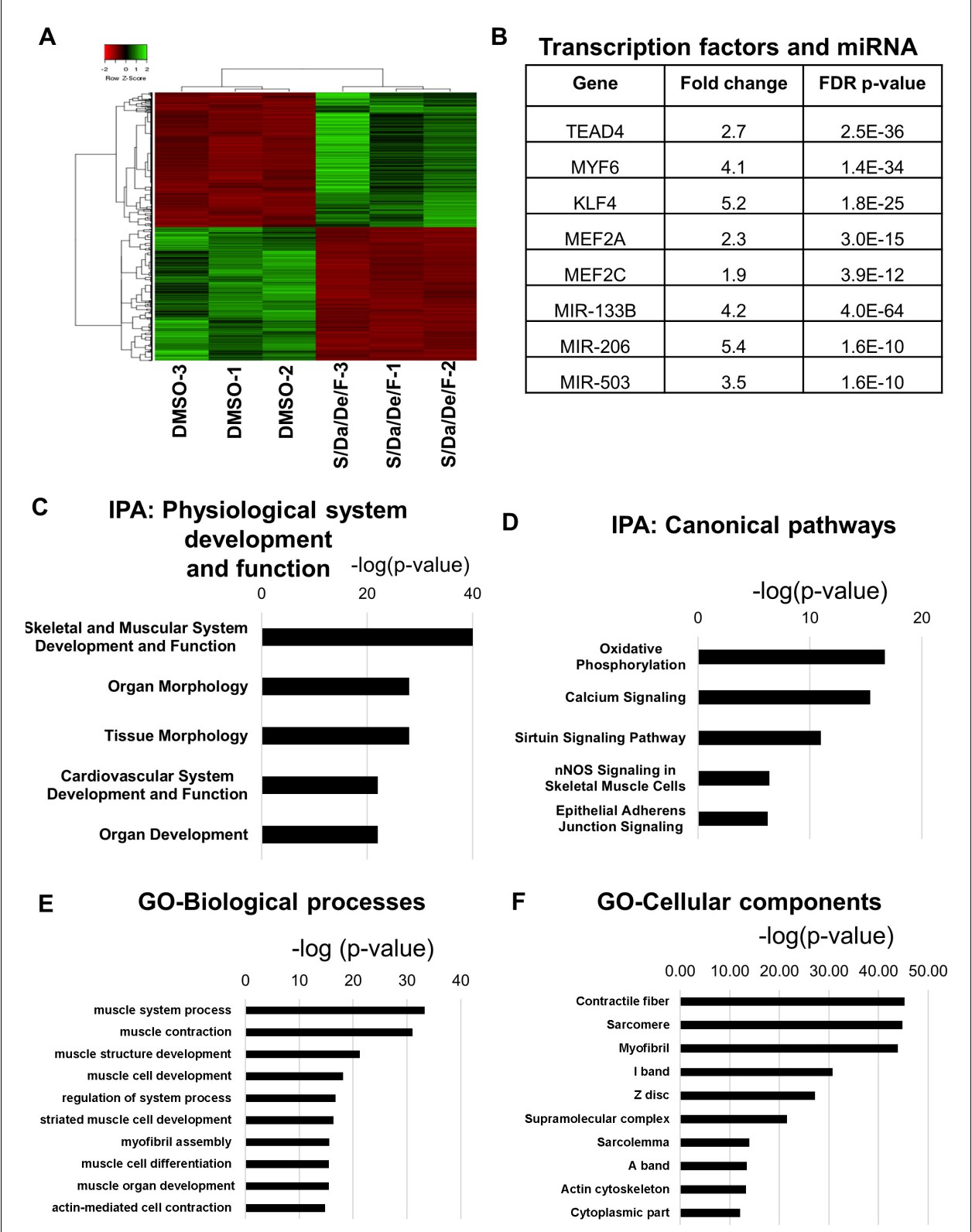

**Figure 6.** Combinatorial treatment induces expression of genes associated with structural maturation. (**A**) Heatmap shows differentially expressed genes in hiPSC-1 myotubes upon combinatorial treatment compared to DMSO from three independent replicates. (**B**) Table shows muscle differentiation associated transcription factors and miRNAs that were upregulated in combinatorial treatment group when compared to DMSO group as revealed by IPA. (**C–D**) Bar graphs show the top physiological systems (**C**) and canonical pathways (**D**) associated with genes upregulated in

*Figure 6 continued on next page*

*Figure 6 continued*

combinatorial treatment group when compared to that of DMSO as revealed by IPA. (**E–F**) Bar graphs show the top biological processes (**E**) and cellular components (**F**) associated with genes upregulated upon combinatorial treatment based on gene ontology (GO) analysis. Data are plotted as –log (p-value) in C-F.

DOI: https://doi.org/10.7554/eLife.47970.017

The following source data and figure supplements are available for figure 6:

**Source data 1.** IPA of upstream regulators of the differentially expressed genes upon combinatorial treatment confirm the pathways targeted by the small molecules.

DOI: https://doi.org/10.7554/eLife.47970.020

**Source data 2.** List shows the targets of transcription factors and miRNA that were found differentially expressed upon combinatorial treatment.

DOI: https://doi.org/10.7554/eLife.47970.021

**Figure supplement 1.** Validation of selected genes revealed by RNA-Sequencing upon combinatorial treatment of PS cell-derived myotubes.

DOI: https://doi.org/10.7554/eLife.47970.018

**Figure supplement 2.** Transcriptomic analysis of genes annotated to loci with increased accessibility following S/Da/De/F treatment.

DOI: https://doi.org/10.7554/eLife.47970.019

---

differentiation and maturation switch occur within 5 days upon switching to differentiation medium and is applicable to MD patient-specific iPS cell-derived myotubes, in addition to unaffected cell lines. We also found that the maturation induced by this treatment is relevant to iPS cell-derived myotubes generated using a transgene-free differentiation protocol, confirming its broad applicability. Our data show that S/Da/De/F treatment is not only significant for the expression of genes associated with maturation in 2D cultures, but it also enhances the contractile capacity of PS cell-derived 3D muscle constructs. This is relevant as the development of organoid-like structures have been shown to provide improved structural organization and, therefore, enhanced tissue functionality, which are critical in the development of reliable platforms for disease modeling and therapy development (*Clevers, 2016*). Taken together, we show that the combinatorial treatment of S/Da/De/F contributes to enhanced myotube differentiation and maturation of PS cell myogenic derivatives, and therefore represents an advance for studying skeletal muscle function and disease in vitro.

# Materials and methods

**Key resources table**

| Reagent type (species) or resource | Designation | Source or reference | Identifiers | Additional information |
|---|---|---|---|---|
| Cell line (*Homo sapiens*, Male) | hiPSC-1 | PMID: 22560081 | PLZ | Control line, available with the Rita Perlingeiro lab |
| Cell line (*Homo sapiens*, Male) | hiPSC-2 | PMID: 26411904 | TC-1133 | Control line, available with RUCDR Infinite Biologics |
| Cell line (*Homo sapiens*, Male) | hiPSC-3 | This study | MNP-120 | Control line, available with the Rita Perlingeiro lab |
| Cell line (*Homo sapiens*, Female) | hiPSC-4 | This study | MNP-119 | Control line, available with the Rita Perlingeiro lab |
| Cell line (*Homo sapiens*, Male) | hESC-1 | WiCell | H9 | ESC control line (WA09) |
| Cell line (*Homo sapiens*, Female) | hESC-2 | WiCell | H1 | ESC control line (WA01) |

*Continued on next page*

Continued

| Reagent type (species) or resource | Designation | Source or reference | Identifiers | Additional information |
|---|---|---|---|---|
| Cell line (*Homo sapiens, Male*) | hiPSC-DMD1 | This study | DMD1108 | DMDΔex31, available with the Rita Perlingeiro lab |
| Cell line (*Homo sapiens, Male*) | hiPSC-DMD2 | PMID: 28658631 | DMD1705 | DMDΔex52-54, available with the Rita Perlingeiro lab |
| Cell line (*Homo sapiens, Male*) | hiPSC-DM1-1 | PMID: 29898953 | DM1-1 | 2,000 CTG repeats in 3'UTR of DMPK gene, available with the Rita Perlingeiro lab |
| Cell line (*Homo sapiens, Male*) | hiPSC-DM1-2 | PMID: 29898953 | DM1-2 | 1,500 CTG repeats in 3'UTR of DMPK gene, available with the Rita Perlingeiro lab |
| Cell line (*Homo sapiens, Female*) | hiPSC-LGMD2A | PMID: 31501033 | 9015 | CAPN3Δex17-24, available with the Rita Perlingeiro lab |
| Chemical compound, drug | Tocriscreen Stem Cell Toolbox | Tocris | Cat# 5060 | 10 µM of each compound |
| Chemical compound, drug | CHIR99021 | Tocris | Cat# 4423 | 10 µM |
| Chemical compound, drug | LDN193189 | Cayman chemical | Cat# 19396 | 200 nM |
| Chemical compound, drug | SB431542 | Cayman chemical | Cat# 13031 | 10 µM |
| Chemical compound, drug | DAPT | Cayman chemical | Cat# 13197 | 10 µM |
| Chemical compound, drug | Dexamethasone | Cayman chemical | Cat# 11015 | 10 µM |
| Chemical compound, drug | Forskolin | Cayman chemical | Cat# 11018 | 10 µM |
| Chemical compound, drug | PD0325901 | Cayman chemical | Cat# 13034 | 10 µM |
| Chemical compound, drug | Doxycycline | Sigma Aldrich | Cat# D9891 | 1 µg/ml |
| Recombinant protein | Recombinant Human FGF-basic | Peprotech | Cat# 100-18B | 5 ng/ml |
| Recombinant protein | Recombinant Human HGF | Stem Cell Technologies | Cat# 78019 | 10 ng/ml |
| Recombinant protein | Recombinant Human IGF-1 | Stem Cell Technologies | Cat# 78022 | 2 ng/ml |
| Commercial assay or kit | iClick EdU Andy Fluor 555 Imaging Kit | GeneCopoeia | Cat# A004 | Cell proliferation assay |
| Antibody | MHC (all isoforms), mouse monoclonal | DSHB | Cat# MF20, RRID: AB_2147781 | Dilution-1:100 (IF) |
| Antibody | Desmin, mouse monoclonal | SCBT | Cat# sc-23879, RRID: AB_627416 | Dilution-1:500 (WB) |
| Antibody | ACTB, mouse monoclonal | SCBT | Cat# sc-4778, RRID: AB_626632 | Dilution- 1:1000 (WB) |
| Antibody | Titin, mouse monoclonal | DSHB | Cat# 9D10, RRID: AB_528491 | Dilution- 1:50 (IF) |

*Continued on next page*

Continued

| Reagent type (species) or resource | Designation | Source or reference | Identifiers | Additional information |
|---|---|---|---|---|
| Antibody | MyHC-neo, mouse monoclonal | DSHB | Cat# N3.36, RRID: AB_528380 | Dilution- 1:50 (IF), 1:200 (WB) |
| Antibody | MyHC-neo, mouse monoclonal | Leica | Cat# MHCN, RRID: AB_563900 | Dilution- 1:20 (IF), 1:200 (WB) |
| Antibody | MyHC-emb, mouse monoclonal | DSHB | Cat# F1.652, RRID: AB_528358 | Dilution- 1:200 (WB) |
| Antibody | MYH1/2, mouse monoclonal | DSHB | Cat# SC-71, RRID: AB_2147165 | Dilution- 1:200 (WB) |
| Antibody | α-actinin, mouse monoclonal | Thermofisher | Cat# MA122863, RRID: AB_557426 | Dilution- 1:25 (IF) |
| Antibody | OCT3/4, mouse monoclonal | SCBT | Cat# C-10, RRID: AB_628051 | Dilution- 1:50 (IF) |
| Antibody | SOX2, goat polyclonal | SCBT | Cat# Y-17, RRID: AB_2286684 | Dilution- 1:50 (IF) |
| Antibody | NANOG, mouse monoclonal | SCBT | Cat# H-2, RRID: AB_10918255 | Dilution- 1:50 (IF) |
| Antibody | SSEA4, mouse monoclonal | SCBT | Cat# sc-21704, RRID: AB_628289 | Dilution- 1:50 (IF) |
| Antibody | Anti-mouse IgG HRP-linked (sheep polyclonal) | GE Healthcare | Cat# NA931, RRID: AB_772210 | Dilution- 1:20000 (WB) |
| Antibody | Alexa fluor 555 goat anti-mouse IgG (goat polyclonal) | Thermofisher | Cat# A-21424, RRID: AB_141780 | Dilution- 1:500 (IF) |
| Other | Alexa Fluor 488 Phalloidin, F-actin probe | Thermofisher | Cat# A12379 | Dilution- 1:40 (IF) |
| Sequence-based reagent | MYH1 | Thermofisher | Assay ID: Hs00428600_m1 | Taqman probe for RT-qPCR |
| Sequence-based reagent | MYH2 | Thermofisher | Assay ID: Hs00430042_m1 | Taqman probe for RT-qPCR |
| Sequence-based reagent | MYH3 | Thermofisher | Assay ID: Hs01074230_m1 | Taqman probe for RT-qPCR |
| Sequence-based reagent | MYH7 | Thermofisher | Assay ID: Hs01110632_m1 | Taqman probe for RT-qPCR |
| Sequence-based reagent | MYH8 | Thermofisher | Assay ID: Hs00267293_m1 | Taqman probe for RT-qPCR |
| Sequence-based reagent | MYOD1 | Thermofisher | Assay ID: Hs02330075_g1 | Taqman probe for RT-qPCR |
| Sequence-based reagent | MYOG | Thermofisher | Assay ID: Hs01072232_m1 | Taqman probe for RT-qPCR |
| Sequence-based reagent | ACTB | Thermofisher | Assay ID: Hs99999903_m1 | Taqman probe for RT-qPCR |
| Sequence-based reagent | GAPDH | Thermofisher | Assay ID: Hs99999905_m1 | Taqman probe for RT-qPCR |
| Sequence-based reagent | SLN | Thermofisher | Assay ID: Hs00161903_m1 | Taqman probe for RT-qPCR |
| Sequence-based reagent | CAPN3 | Thermofisher | Assay ID: Hs01115989_m1 | Taqman probe for RT-qPCR |
| Sequence-based reagent | ATP2A1 | Thermofisher | Assay ID: Hs01115989_m1 | Taqman probe for RT-qPCR |
| Sequence-based reagent | ENO3 | Thermofisher | Assay ID: Hs01093275_m1 | Taqman probe for RT-qPCR |

*Continued*

| Reagent type (species) or resource | Designation | Source or reference | Identifiers | Additional information |
|---|---|---|---|---|
| Sequence-based reagent | MYF6 | Thermofisher | Assay ID: Hs00231165_m1 | Taqman probe for RT-qPCR |
| Sequence-based reagent | CKM | Thermofisher | Assay ID: Hs00176490_m1 | Taqman probe for RT-qPCR |
| Sequence-based reagent | KLF4 | Thermofisher | Assay ID: Hs01034973_g1 | Taqman probe for RT-qPCR |
| Sequence-based reagent | TNNT3 | Thermofisher | Assay ID: Hs00952980_m1 | Taqman probe for RT-qPCR |
| Sequence-based reagent | CDH11 | Thermofisher | Assay ID: Hs00901479_m1 | Taqman probe for RT-qPCR |
| Sequence-based reagent | EYA2 | Thermofisher | Assay ID: Hs00193347_m1 | Taqman probe for RT-qPCR |
| Sequence-based reagent | FST | Thermofisher | Assay ID: Hs01121165_g1 | Taqman probe for RT-qPCR |
| Sequence-based reagent | CEBPB | Thermofisher | Assay ID: Hs00270923_s1 | Taqman probe for RT-qPCR |
| Sequence-based reagent | CEBPD | Thermofisher | Assay ID: Hs00270931_s1 | Taqman probe for RT-qPCR |
| Sequence-based reagent | FKBP5 | Thermofisher | Assay ID: Hs01561006_m1 | Taqman probe for RT-qPCR |
| Sequence-based reagent | NOTCH2 | Thermofisher | Assay ID: Hs01050702_m1 | Taqman probe for RT-qPCR |
| Sequence-based reagent | HES1 | Thermofisher | Assay ID: Hs00172878_m1 | Taqman probe for RT-qPCR |
| Sequence-based reagent | JAG1 | Thermofisher | Assay ID: Hs01070032_m1 | Taqman probe for RT-qPCR |
| Sequence-based reagent | COL1A1 | Thermofisher | Assay ID: Hs00164004_m1 | Taqman probe for RT-qPCR |
| Sequence-based reagent | ID3 | Thermofisher | Assay ID: Hs00954037_g1 | Taqman probe for RT-qPCR |
| Sequence-based reagent | SERPINE1 | Thermofisher | Assay ID: Hs00167155_m1 | Taqman probe for RT-qPCR |
| Sequence-based reagent | PPARGC1A | Thermofisher | Assay ID: Hs00173304_m1 | Taqman probe for RT-qPCR |
| Sequence-based reagent | RGS2 | Thermofisher | Assay ID: Hs01009070_g1 | Taqman probe for RT-qPCR |
| Sequence-based reagent | NR4A1 | Thermofisher | Assay ID: Hs00374226_m1 | Taqman probe for RT-qPCR |

## Cell lines

Studies involved de-identified fibroblast samples according to procedures approved by the Institutional Review Board of the University of Minnesota (Ref: 0904M63241). PS cell lines used in this study are described in key resources table. hiPSC-3, hiPSC-4 and hiPSC-DMD1 were generated by the Pluripotent Stem Cell Facility at Cincinnati Children's Hospital Medical Center. Pluripotency characterization is shown in *Figure 3—figure supplement 1*. All muscular dystrophy patient iPS cell lines were authenticated by verification of genetic mutation by southern blot or sanger sequencing. All tested iPS cell lines were negative for mycoplasma contamination.

## Mice

Animal experiments were carried out according to protocols (protocol ID: 1702-34580A) approved by the University of Minnesota Institutional Animal Care and Use Committee. Teratoma studies were performed by injecting $1.5 \times 10^6$ PS cells in the quadriceps of NSG mice (from Jackson labs). Before

injection, cells were resuspended in 1:1 solution DMEM-F12 and Matrigel (final volume including cells: 65 μl).

## Cell culture and myogenic differentiation of PS cells

ES and iPS cells were maintained in mTeSR1 medium (Stem Cell Technologies) on Matrigel-coated plates. ES/iPS cells were dissociated with Accumax and passaged once they reached 90% confluency and plated with 10 μM ROCK inhibitor, Y-27632 (APExBIO). To induce conditional expression of PAX7, ES/iPS cells were co-transduced with lentiviral vectors, pSAM2-iPAX7-IRES-GFP and FUGW-rtTA to generate iPAX7 cells (*Darabi et al., 2012*). For myogenic differentiation, $1 \times 10^6$ iPAX7-ES/iPS cells were plated with 10 μM Y-27632 in a 60 mm petri dish and incubated for 2 days on a shaker at 60 rpm at 37°C to derive embryoid bodies (EBs). Cells were then switched to myogenic medium (MM) supplemented with 10 μM GSK3β inhibitor (CHIR99021; Tocris). Myogenic medium consisted of IMDM basal medium (Gibco) supplemented with 15% fetal bovine serum, 10% horse serum, 1% penicillin/streptomycin (Invitrogen), 1% glutamax (Gibco), 1% KnockOut Serum Replacement (KOSR; Gibco), 50 μg/ml ascorbic acid (Sigma-Aldrich), 4.5 mM monothioglycerol (MP biomedicals). After 2 days, cells were switched to MM supplemented with 200 nM BMP inhibitor (LDN193189; Cayman Chemical) and 10 μM TGFβ inhibitor (SB431542, Cayman Chemical). 24 hr later, on day 5, 1 μg/ml Doxycycline (Dox; Sigma-Aldrich) was added to the medium. After 24 hr, cells were switched to fresh MM with 1 μg/ml Dox and incubated at 37°C for 2 days. EBs were collected and 1/10<sup>th</sup> of the EBs volume was plated on gelatin coated T75 flasks in expansion medium (EM), consisting of MM with 1 μg/ml Dox and 5 ng/ml human basic fibroblast growth factor (bFGF; PeproTech). After 4 days, cells were dissociated with 0.25% trypsin-EDTA and FACS sorted for GFP to derive myogenic progenitors which were plated at a density of $2 \times 10^6$/T75 flask in EM. Myogenic progenitors were passaged upon reaching 90% confluence with 0.25% trypsin-EDTA at a ratio of 1:6 to 1:8. To induce terminal differentiation, myogenic progenitors at P3-4 were plated at a density of 75,000 cells/well of a 24-well plate and allowed to grow confluent for 3 days. The medium was then switched to low nutrient differentiation medium (DM) consisting of DMEM-KO supplemented with 20% KOSR, 1% Non-Essential amino acids (NEAA; Gibco), 1% glutamax and 1% penicillin-streptomycin and incubated for 5 days to derive myotubes.

Transgene-free myogenic differentiation was performed following a previously described protocol (*Xi et al., 2017*). Briefly, hiPS cells were plated as single cells onto a well of a Matrigel-coated 6-well dish in mTeSR1 with 10 μM Y-27632 (day 0). The next day (day 1), medium was switched to basal medium (1% ITS-G and 1% penicillin/streptomycin in DMEM/F12) supplemented with 3 μM CHIR99021. At day 3, medium was switched to basal medium supplemented with 200 nM LDN193189 and 10 μM SB431542. At day 5, medium was switched to basal medium supplemented with 3 μM CHIR99021 and 20 ng/ml bFGF. At day 7, medium was switched to 15% KOSR and 1% penicillin/streptomycin in DMEM supplemented with 10 ng/ml HGF and 2 ng/ml IGF1 for 14 days. Medium was replaced every other day. At day 21, cells were dissociated with Collagenase IV for 5 min followed by TryPLE Express for 5 min. Cell suspension was filtered through 100 μm and 40 μm cell strainers, sequentially. Cells were cultured and expanded on Matrigel coated dishes in MM. For terminal myotube differentiation, cells were grown to confluency and then MM was replaced by DM.

## Small molecule library screening

Small molecule library screening was performed using the Tocriscreen Stem Cell Toolbox Kit (Tocris), which consists of 80 compounds. Myogenic progenitors were seeded at 12,000 cells/well in 96-well plates (triplicates) and incubated at 37°C for 3 days to reach 100% confluency. After 3 days, the cells were switched to DM supplemented with 10 μM of the compounds from the library or with DMSO. Each well contained an individual compound. Cells were incubated in this differentiation medium for 5 days to derive myotubes. Cells were then fixed and processed for immunostaining of MyHC and counterstained with DAPI. Stained plates were imaged such that most of the well area was covered. The ratio of area of MyHC/DAPI staining was quantified using Image J and normalized to that of DMSO treated cells. Compounds showing a significant increase of 1.2-fold or more in ratio of MyHC/DAPI relative to DMSO among replicates and among the three cell lines tested were used for further studies: S (Cayman Chemical), Da (Selleckchem), De (Cayman Chemical), P (Cayman

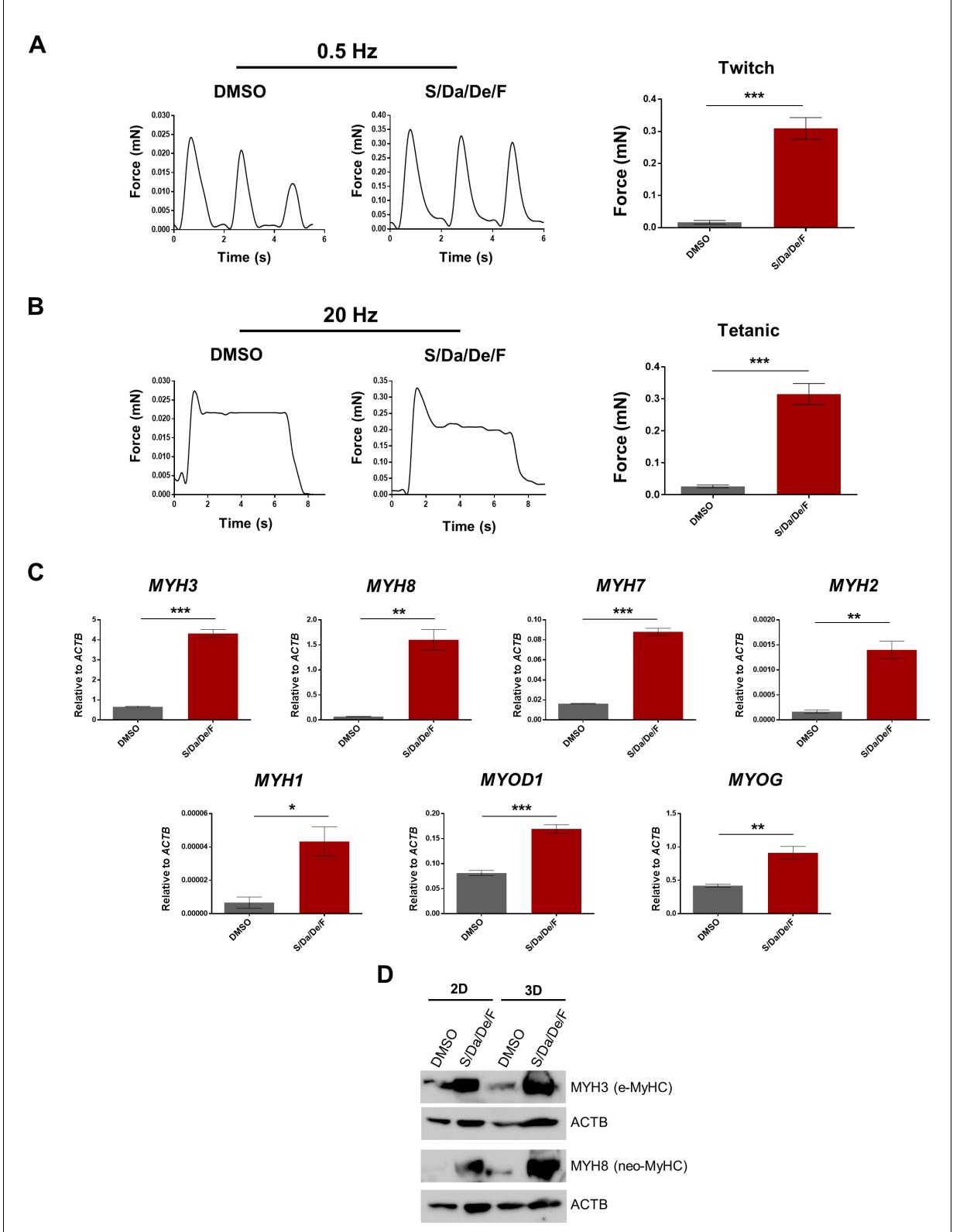

**Figure 7.** Increased contractile force generation in PS cell-derived 3D muscle constructs upon combinatorial treatment. (**A–B**) Representative twitch (**A**) and tetanic (**B**) force patterns at 0.5 Hz and 20 Hz, respectively, generated by hiPSC-1 3D muscle constructs differentiated with combinatorial treatment or DMSO. Bar graphs show the twitch force (**A**) and tetanic force (**B**) as mean of three independent replicates ± S.E.M. ***p<0.001. Nine twitch peaks and three tetanic measurements from three independent muscle constructs were used for analysis. (**C**) Bar graphs show myogenic genes expression

*Figure 7 continued on next page*

*Figure 7 continued*

analysis relative to *ACTB* in 3D muscle constructs differentiated with combinatorial treatment or DMSO (from A and B). Data are shown as mean of three independent replicates ± S.E.M. *p<0.05 **p<0.01 ***p<0.001. (D) Protein expression analysis of MYH3 (e-MyHC) and MYH8 (neo-MyHC) by western blot of hiPSC-1 3D muscle constructs and 2D differentiated myotubes with combinatorial treatment or DMSO. Actin is shown as loading control.

DOI: https://doi.org/10.7554/eLife.47970.022

The following figure supplement is available for figure 7:

**Figure supplement 1.** Differentiation of PS cell-derived 3D muscle constructs in the presence of S/Da/De/F enhances the expression of adult *MYH* isoforms.

DOI: https://doi.org/10.7554/eLife.47970.023

Chemical) and F (Cayman Chemical. Different combinations were made with selected compounds and their effect on differentiation was tested and quantified as above.

## Immunofluorescence staining

For immunofluorescence staining, samples were fixed with 4% PFA for 20–30 min at RT, followed by permeabilization with 0.3% Triton X-100 in PBS for 20 min at RT. Samples were then blocked with 3% BSA in PBS for 1 hr. Primary antibody diluted in 3% BSA was added after blocking and was incubated overnight at 4°C. Following this incubation, samples were washed with PBS and incubated with secondary antibody and DAPI for 45–60 min at RT in dark. After this incubation, samples were washed with PBS and stored in dark at 4°C until imaging. For immunofluorescence staining of 3D muscle constructs, these were fixed in 4% PFA for 1 hr at RT, followed by incubation in a solution of 30% Sucrose and 5% DMSO for 4 hr at RT and then incubated in a solution of 15% Sucrose, 2.5% DMSO and 50% OCT embedding medium at 4°C overnight. Next, constructs were embedded in OCT and frozen with liquid nitrogen cooled isopentane. Frozen 3D constructs were cryosectioned longitudinally at 45 μm thickness. Frozen sections were allowed to dry at RT for 15 min and the sections were rehydrated in PBS for 5 min prior to 10 min fixation with 4% PFA. Following staining procedure, slides were mounted with coverslips using ProLong Gold Antifade Mountant with DAPI (Invitrogen). The stained sections were imaged using Zeiss upright microscope.

## Myotubes fusion index

Differentiated myotubes were processed for pan-MyHC immunostaining. Cells were imaged using Zeiss upright microscope and analysis was performed from aleatory fields of the well. Myotubes were identified as elongated MyHC (+) cells containing at least two nuclei. Fusion index was calculated as the percentage of nuclei within myotubes relative to the total number of nuclei. Approximately 500 nuclei and 100 myotubes were counted for fusion index and number of nuclei per myotube, respectively, for each replicate.

## EdU cell proliferation assay

This was performed using iClick EdU Andy Fluor 555 Imaging Kit (GeneCopoeia). Cells were incubated with 10 μM of EdU for 24 hr. After this incubation, cells were fixed with 4% PFA for 15 min at RT and then permeabilized with 0.3% Triton X-100 for 20 min at RT. Following this, EdU staining was performed as per the manufacturer's instructions. Stained cells were imaged using Cytation 3 (Bio-Tek) and EdU positive nuclei were quantified using Image J.

## Western blotting

Protein extraction was performed using lysis buffer (20 mM Tris HCl, 0.1 mM EDTA, 1 mM DTT, 20 μg/ml soybean trypsin inhibitor, 28 μM E64 and 2 mM PMSF) combined with 1X Laemmli sample buffer. Samples were boiled at 95°C for 10 min. The total protein concentration of the lysate was quantified using Bradford assay. 100 μg of total protein was electrophoresed in 7.5% SDS-PAGE gel. The proteins were transferred onto immobilon PVDF membrane (Millipore). The blot was blocked with 5% dry milk in TBST for 1 hr at RT. After blocking, the blot was incubated with primary antibody diluted in 5% BSA in TBST overnight at 4°C. Following this incubation, the blot was washed three times with TBST and then incubated with HRP conjugated secondary antibody. After three

washes with TBST, protein detection was performed using Pierce ECL or Supersignal west chemiluminescent substrate (Thermo Fisher Scientific). The chemiluminescence signal was captured in X-ray film or using chemidoc imager (Bio-Rad Laboratories).

## Antibodies

The following antibodies were used for immunofluorescence or western blot: pan-MyHC (MF20; DSHB), Titin (9D10; DSHB), MyHC-neo (N3.36; DSHB) (MHCN; Leica), MYH3 (F1.652; DSHB), MYH1/2 (SC-71; DSHB), α-actinin (MA122863; Thermofisher), Desmin (sc-23879; SCBT), ACTB (sc-4778; SCBT), OCT3/4 (C-10; SCBT), SOX2 (Y-17; SCBT), NANOG (H-2; SCBT), Alexa Fluor 488 Phalloidin (A12379; Thermofisher), Alexa fluor 555 goat anti-mouse IgG (A-21424; Thermo Fisher), Alexa fluor 488 goat anti-rabbit IgG (A-11008; Thermo Fisher), and mouse IgG HRP-linked (NA931; GE Healthcare).

## RNA isolation and quantitative RT-PCR

Cells were lysed using Trizol reagent (Thermo Fisher) and RNA was extracted using purelink RNA mini kit (Thermo Fisher) with on-column DNAse treatment following manufacturer's instructions. RNA concentration was quantified using Nanodrop. For quantitative RT-PCR analysis, reverse transcription was performed using Superscript Vilo cDNA synthesis kit (Thermo Fisher) as per manufacturer's instruction. qPCR was performed using taqman probes (Applied Biosystems) and Premix Ex Taq probe qPCR kit (Takara). For each qPCR reaction in 384-well plate, cDNA amount corresponding to 10 ng of total RNA, 0.5 µl of taqman probe and 5 µl of 2X master mix was utilized. QPCR was performed using QuantStudio 6 Flex Real-Time PCR System and the $C_t$ values were determined. $C_t$ value for gene of interest was normalized to that of the house keeping control using the $2^-$delta $C_t$ calculation and compared between the treated and untreated groups. Following are the taqman probes used in this study, *MYH1* (Hs00428600_m1), *MYH2* (Hs00430042_m1), *MYH3* (Hs01074230_m1), *MYH7* (Hs01110632_m1), *MYH8* (Hs00267293_m1), *MYOG* (Hs01072232_m1), *MYOD1* (Hs02330075_g1), *ACTB* (Hs99999903_m1), *GAPDH* (Hs99999905_m1), *SLN* (Hs00161903_m1), *CAPN3* (Hs01115989_m1), *ATP2A1* (Hs01115989_m1), *ENO3* (Hs01093275_m1), *MYF6* (Hs00231165_m1), *CKM* (Hs00176490_m1), *KLF4* (Hs01034973_g1), *TNNT3* (Hs00952980_m1), *CDH11* (Hs00901479_m1), *EYA2* (Hs00193347_m1), *FST* (Hs01121165_g1), *CEBPB* (Hs00270923_s1), *CEBPD* (Hs00270931_s1), *FKBP5* (Hs01561006_m1), *NOTCH2* (Hs01050702_m1), *HES1* (Hs00172878_m1), *JAG1* (Hs01070032_m1), *COL1A1* (Hs00164004_m1), *ID3* (Hs00954037_g1), *SERPINE1* (Hs00167155_m1), *PPARGC1A* (Hs00173304_m1), *RGS2* (Hs01009070_g1), *NR4A1* (Hs00374226_m1).

## Transmission electron microscopy (TEM)

Transmission electron microscopy (TEM) was performed on myotubes cultured in 6-well plates and differentiated for 19 days under standard conditions or in the presence of small molecules. Cells were pre-fixed at room temperature for one hour in 2% (w/v) paraformaldehyde 2% (w/v) glutaraldehyde in PBS, then rinsed and stored in PBS for a few days. Samples were prepared by post-fixation in 1% (w/v) OsO4 in PBS, followed by gradual dehydration in ethanol that includes a staining with step with 1% (w/v) uranyl acetate in 70° ethanol and embedding in Epon resin (EMS, Fort Washington, PA, USA). Blocks were cut in 70 nm ultrathin sections which were further stained with uranyl acetate and lead citrate. Sections were examined using a Philips CM120 electron microscope (Philips, Eindhoven, Netherlands) operated at 80kV, and photographed with SIS Morada digital camera (Olympus, Münster, Germany). Experiments were carried out in triplicates and in a blind manner.

## ATAC-seq

Analysis of chromatin accessibility was performed following the protocol described by Buenrostro and colleagues (*Buenrostro et al., 2015*). Briefly, 50,000 cells from 2 day DMSO- and S/Da/De/F-treated cultures were collected using trypsin/EDTA and washed with cold PBS prior to permeabilization and Tn5-mediated transposition for 30 min at 37°C. Following DNA extraction using MinElute Qiagen columns, transposed DNA was used for primer extension and 5 cycles of PCR amplification to insert Illumina-compatible adapter-barcodes. Final number of cycles required for library amplification was determined by qPCR. Libraries were purified using AMPure beads and then resuspended in

$H_2O$. Libraries were sequenced at the University of Minnesota Genomic Center on a lane of the NextSeq 550 in paired-end mode at an average depth of 40M reads/sample.

Reads were then mapped to the human genome (hg38) using bowtie2 (*Langmead and Salzberg, 2012*) (parameters `-I 25 -X 2000 –local –dovetail –no-mixed –no-discordant`) and filtered to remove PCR duplicates. Peaks were identified using MACS2.1 (*Zhang et al., 2008*) (parameters –nomodel –nolambda –keep-dup all –call-subpeaks) and then analyzed using BEDtools (*Quinlan and Hall, 2010*) to identify treatment-specific peaks. For this analysis, we considered peaks detected in 2 of 3 biological replicates. Analysis of differential accessibility was performed by generating a list of peaks representative of all samples using the peak summits identified by MACS2. Each summit was extended 50 bp in both directions and the resulting lists of peaks were combined, sorted and merged to obtain a dataset of unique and non-overlapping loci. This list was then used to extrapolate the sequencing depth coverage from each sample bedgraph file. Coverage files were then analyzed using DEseq2 (*Love et al., 2014*) to identify loci with differential chromatin accessibility. For the subsequent analyses, loci were converted to the hg19 human genome release. Annotation of the S/Da/De/F specific peaks was performed using GREAT (*McLean et al., 2010*) using a two gene association and 500 kb regulatory domain. Analyses included proximal promoters and distal enhancers. GO analysis was performed using DAVID (*Huang et al., 2009*). Enriched motifs were identified in a region of ±200 bp from the peak center using MEME-ChIP (*Machanick and Bailey, 2011*). Files for visualization using IGV (*Thorvaldsdóttir et al., 2013*) were generated by converting. wig files to the bigwig format.

## RNA sequencing

Myotubes treated with DMSO or S/Da/De/F differentiated for 5 days were used for RNA sequencing in triplicates. RNA was isolated as described above. 500 ng of total RNA was used for generating dual-indexed libraries using the TruSeq stranded mRNA library kit. The libraries were sequenced in NextSeq 550 sequencer using 75 bp paired end run at around 20 million reads per sample. 75 bp FastQ paired-end reads (n = 20.9 Million per sample) were trimmed using Trimmomatic (v 0.33). Quality control on raw sequence data was performed with FastQC. Reads were mapped to the human genome (hg38) reference using Hisat2 (v2.1.0). Gene quantification was done via Cuffquant for FPKM values and Feature Counts for raw read counts. Differentially expressed genes were identified using the edgeR (negative binomial) feature in CLC genomics work bench (Qiagen) using raw read counts. We filtered the generated list based on a minimum 2X Absolute Fold Change and FDR corrected p<0.05. These filtered genes were then imported into Ingenuity Pathway Analysis Software (Qiagen) for identification of canonical pathways, upstream regulators and their targets. Gene ontology enrichment analysis was performed by uploading the list of differentially expressed genes in the GO database to identify the top enriched cellular components and biological processes. Heatmap of differentially expressed genes was generated using the heatmapper web tool as per the manufacturer's instructions.

## Three-dimensional muscle construct generation and contractile force measurement

Three-dimensional muscle constructs were generated in home-made culture wells having dimensions of 15 mm ×5 mm × 5 mm (length ×width × depth), with each well containing two posts near the ends of the well. A suspension of hiPS cell-derived myogenic progenitors (10 million cells/mL, passage 4) was prepared in a solution containing 6 mg mL$^{-1}$ bovine fibrinogen (Sigma-Aldrich), 1-unit bovine thrombin (Sigma-Aldrich), and 10% (v/v) growth factor reduced Matrigel (R and D), and the suspension was quickly pipetted into the culture wells, followed by gelation at 37°C for 1 hr. All the constructs were cultured in the myogenic expansion medium for 3 days, followed by differentiation for 5 days in the KOSR medium supplemented with S/Da/De/F (each dissolved in DMSO at 10 μM). Controls in which the KOSR differentiation medium containing the same amount of DMSO (0.4% v/v) were conducted. On day 3 of differentiation, 25% of the medium was replaced with the fresh medium. All the media were supplemented with 2 mg/mL ε-aminocaproic acid (Sigma-Aldrich) to prevent fibrin degradation.

Contractile forces generated by the constructs in response to electrical stimulation at 0.5 Hz (twitch) or 20 Hz (tetanus) were measured on a custom-built apparatus after five days of

differentiation (*Black et al., 2009*). In brief, a construct was maintained at 37°C, and the two ends were mounted on two pins, one of which was adjustable and connected to a force transducer (Harvard Apparatus, 60–2994 model). Prior to measurements, the construct was stretched by 20% of its initial length with the adjustable pin. Electrical pulses were generated with a cardiac stimulator (Astro-Med Inc, S88 × Model, 10 ms pulse width) at a frequency of 0.5 or 20 Hz for 6 s. Contractile forces were recorded using LabView and the data were analyzed using MATLAB (the code is available at GITHUB, https://github.com/weishenumn/contractile-force-analysis.git ; *Shen, 2019*; copy archived at https://github.com/elifesciences-publications/contractile-force-analysis) (*Black et al., 2009*). For each twitch peak or tetanic plateau, the maximum force was used for analysis. The construct was then treated with TRI Reagent (Sigma-Aldrich) for qPCR analysis or immediately frozen in liquid nitrogen for Western blotting. For each differentiation condition, three constructs were quantitatively examined.

## Statistics

Differences between samples were assessed by using the Student's two-tailed t test for independent samples or two-way ANOVA for multiple comparisons. Statistical analyses were performed using Prism Software (GraphPad).

## Acknowledgements

This project was supported by funds from the NIH, grants R01 AR055299 and AR071439 (RCRP) and the NSF CAREER DMR-1151529 (WS). RM-G was funded by CONACyT-Mexico (#394378). We also thank the generous support from ADVault, Inc and MyDirectives.com (RCRP). The cytogenetic analyses were performed in the Cytogenomics Shared Resource at the University of Minnesota with support from the comprehensive Masonic Cancer Center NIH Grant #P30 CA077598-09. The monoclonal antibodies to MyHC, Titin, MyHC-neo and MYH1/2 were obtained from the Developmental Studies Hybridoma Bank developed under the auspices of the NICHD and maintained by the University of Iowa. We thank Prof. Robert T Tranquillo, Dr. Jeremy Schaefer, and Dr. Lauren Black for the use of the contractile force measurement apparatus and for facilitating data analysis.

## Additional information

### Competing interests

Fabrizio Rinaldi, Joy Aho: is affiliated with Bio-Techne. The author has no financial interests to declare. The other authors declare that no competing interests exist.

### Funding

| Funder | Grant reference number | Author |
| --- | --- | --- |
| National Institutes of Health | R01 AR055299 | Rita CR Perlingeiro |
| National Institutes of Health | R01 AR071439 | Rita CR Perlingeiro |
| National Science Foundation | CAREER DMR-1151529 | Wei Shen |
| Consejo Nacional de Ciencia y Tecnología | 394378 | Ricardo Mondragon-Gonzalez |
| ADVault, Inc | | Rita CR Perlingeiro |
| MyDirectives.com | | Rita CR Perlingeiro |

The funders had no role in study design, data collection and interpretation, or the decision to submit the work for publication.

### Author contributions

Sridhar Selvaraj, Jeanne Lainé, Conceptualization, Validation, Investigation, Visualization, Methodology, Writing—review; Ricardo Mondragon-Gonzalez, Conceptualization, Formal analysis, Validation, Investigation, Visualization, Methodology, Writing—original draft, Writing—review and

editing; Bin Xu, Conceptualization, Formal analysis, Validation, Investigation, Methodology, Writing—original draft; Alessandro Magli, Conceptualization, Software, Formal analysis, Investigation, Visualization, Methodology, Writing—original draft; Hyunkee Kim, James Kiley, Formal analysis, Validation, Investigation, Visualization; Holly Mckee, Formal analysis, Investigation; Fabrizio Rinaldi, Joy Aho, Resources; Nacira Tabti, Conceptualization, Writing—original draft, Project administration; Wei Shen, Conceptualization, Supervision, Methodology, Writing—original draft, Project administration; Rita CR Perlingeiro, Conceptualization, Supervision, Funding acquisition, Writing—original draft, Project administration, Writing—review and editing

### Author ORCIDs
Sridhar Selvaraj https://orcid.org/0000-0002-9736-8391
Ricardo Mondragon-Gonzalez https://orcid.org/0000-0001-5645-3090
Alessandro Magli http://orcid.org/0000-0003-3874-2838
Rita CR Perlingeiro https://orcid.org/0000-0001-9412-1118

### Ethics
Human subjects: Studies involved de-identified fibroblast samples according to procedures approved by the Institutional Review Board of the University of Minnesota (Ref: 0904M63241). Informed consent, and consent to publish was obtained.
Animal experimentation: Animal studies were carried out according to protocols (protocol ID: 1702-34580A) approved by the University of Minnesota Institutional Animal Care and Use Community.

### Decision letter and Author response
Decision letter https://doi.org/10.7554/eLife.47970.029
Author response https://doi.org/10.7554/eLife.47970.030

## Additional files

### Supplementary files
• Transparent reporting form DOI: https://doi.org/10.7554/eLife.47970.024

### Data availability
Sequencing data have been deposited in GEO under accession code: GSE130592.

The following dataset was generated:

| Author(s) | Year | Dataset title | Dataset URL | Database and Identifier |
|---|---|---|---|---|
| Sridhar S, Ricardo M, Alessandro M, Rita P | 2019 | Combinatorial small molecule treatment enhances the in vitro maturation of pluripotent stem cell-derived myotubes | https://www.ncbi.nlm.nih.gov/geo/query/acc.cgi?acc=GSE130592 | NCBI Gene Expression Omnibus, GSE130592 |

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
