## [Decision Letter]

**Acceptance summary:**

This is an important study that addresses a major bottleneck in the field, the development of approaches to improve maturation of ES- and iPS-derived myogenic cells in vitro, to provide a novel platform for disease modeling and drug screening.

**Decision letter after peer review:**

Thank you for submitting your article "Screening identifies small molecules that enhance the maturation of human pluripotent stem cell-derived myotubes" for consideration by *eLife*. Your article has been reviewed by three peer reviewers, and the evaluation has been overseen by Didier Stainier as the Senior Editor. The following individuals involved in review of your submission have agreed to reveal their identity: Bradley B Olwin (Reviewer #2); Pier Lorenzo Puri (Reviewer #3).

The reviewers have discussed the reviews with one another and the Reviewing Editor has drafted this decision to help you prepare a revised submission.

While the reviewers find the work of significant interest, there are some issues that should be addressed before acceptance, mainly related to myotube maturation and to the cellular mechanism affected by the compound combination, listed below.

Essential revisions:

1) While the upregulation of fetal and adult myogenic genes is clearly presented, it would be useful to show and discuss the downregulation of embryonic muscle genes by RNA-seq and ATAC-seq, in order to further strengthen the interpretation of a switch in developmental stage. As relevant controls, inclusion of human embryonic and adult skeletal muscle should be included. Alternatively, a comparison of the presented RNA-seq and ATAC-seq data with available human embryonic and adult gene expression should be performed, in order to evaluate the extent of myotube maturation generated this novel protocol compared to embryonic and adult tissues.

2) While expression of mature forms of MyHCs is some evidence for enhancing maturity, the authors never provided data that the mature MyHCs are translated and present in sarcomeric-like structures within the differentiated myotubes. The only protein data provided were for neonatal MyHC, which is expressed perinatally but is not a mature myosin. Complicating the interpretation is the fact that the antibody used recognizes adult fast and perinatal myosin. It is absolutely critical that the authors identify and quantify the "mature" myosins that are translated and present in contractile structures as this is the predominant discovery of the manuscript. Translation of mature MyHC isoforms and insertion into the relevant cellular structures is bona fide evidence of maturity. Indeed, the degree of maturity might be quantitatively measured from a ratio of the protein present for immature vs. mature myosin isoforms.

3) The inhibitors and one activator act on signaling pathways but the authors did not provide evidence for whether the pathways targeted were affected. Moreover, it would be useful to know whether the combinatorial effect was substantially different than individual factors or lesser combinations. These data might provide better knowledge regarding the mechanisms involved as opposed to much further downstream analysis of transcriptomics and epigenetics.

4) In the protocol for HTS, the increased ratio of MyHC (+)/DAPI was used as readout to screen for compounds that could improve maturation of PS-derived myotubes. The author's rationale is that statistically significant increase in MyHC/DAPI area ratio in myotubes reflects increased maturation. This approach revealed to be correct, as retrospectively the strategy led to the selection of compounds that actually improve maturation; however, the lack of some details on the experimental protocol does not help on the mechanistic interpretation of the data. For instance, the authors state that the small molecules were added after cells were cultured for three days in the presence of expansion medium (bFGF and Dox), and in concomitance with the switch to differentiation medium, and MyHC expression was evaluated 5 days later. There is no information on whether cells were already differentiated or were proliferating at the time of the beginning of exposure to small molecules. Thus, it remains unclear whether the increased MyHC/DAPI area ratio is due to an effect on cell proliferation, myoblast fusion and/or transcriptional activation of specific genes in post-mitotic myotubes. This also precludes a functional interpretation of the mechanism by which the combined effect of TGFβ, MEK and γ-Secretase inhibition, adenylyl cyclase activation and glucocorticoids exerts the observed effect. As this information seems crucial to further optimize this approach, I encourage the authors to test the effects of these small molecules at sequential stages of PS-derived formation of skeletal muscles, while monitoring cell proliferation, cell fusion and multinucleation with a time course from the beginning of exposure through the whole time of treatment with small molecules.

5) Within the same line of point #1, the authors have tested the small molecules in combination (or by subtraction of each of them) always within the same protocol (e.g. exposure at the time of medium switch). It is unknown what is the differentiation status of the cells at that timing, but presumably they are not synchronous, but are rather quite heterogeneous in term of proliferation/differentiation stages. As each of these molecules seems to affect different processes along the sequential stages of skeletal myogenesis, it is highly possible that their optimal synergistic effect can be appreciated by protocols of sequential exposure to each of them, likely with some overlaps of exposures, rather than by an empirical treatment with simultaneous exposure to all compounds, as reported here. The authors should therefore consider evaluating the effect of protocols of sequential exposure in various combinations as compared to simultaneous exposure. As this approach will require multiple experimental points, I would recommend reducing the burden of work by taking one representative endpoint easy to detect – e.g. one representative *MYH* isoform, as *MYH8*.

6) More details on the methods used for generating myogenic progenitors in transgene-free conditions should be provided.

7) The 3D culture data are not sufficiently developed. Are these myotubes more or less mature than the 2D cultures? The Western blot in Figure 6D is relevant for the last two issues raised above as it is clear that neonatal MyHC is a minority of the total MyHC present. This aspect of the work should be further developed or not included as aside from the force measurements, it adds little to demonstrating maturity of the myotubes or aiding in elucidation of the mechanisms involved.

---

## [Author Response]

Essential revisions:1) While the upregulation of fetal and adult myogenic genes is clearly presented, it would be useful to show and discuss the downregulation of embryonic muscle genes by RNA-seq and ATAC-seq, in order to further strengthen the interpretation of a switch in developmental stage. As relevant controls, inclusion of human embryonic and adult skeletal muscle should be included. Alternatively, a comparison of the presented RNA-seq and ATAC-seq data with available human embryonic and adult gene expression should be performed, in order to evaluate the extent of myotube maturation generated this novel protocol compared to embryonic and adult tissues.

We thank the reviewers for this suggestion. Since we do not have access to human embryonic and adult skeletal muscle tissue samples, we investigated a selected set of embryonic genes based on a previous study that compared the gene expression profile of human embryonic vs. fetal myoblast-derived myotubes (Biressi, Tagliafico, et al., 2007). We have now included qPCR data showing that genes associated with embryonic myotubes, including *MEOX1, PAX3, CDH11, EYA2* and *FST*, are significantly downregulated upon treatment with small molecules (Figure 6—figure supplement 1B of revised manuscript). Since these genes were reported to be expressed at higher levels in embryonic when compared to fetal myotubes (Biressi, Tagliafico, et al., 2007), downregulation of these genes upon small molecule treatment corroborates our findings suggesting developmental stage switch.

2) While expression of mature forms of MyHCs is some evidence for enhancing maturity, the authors never provided data that the mature MyHCs are translated and present in sarcomeric-like structures within the differentiated myotubes. The only protein data provided were for neonatal MyHC, which is expressed perinatally but is not a mature myosin. Complicating the interpretation is the fact that the antibody used recognizes adult fast and perinatal myosin. It is absolutely critical that the authors identify and quantify the "mature" myosins that are translated and present in contractile structures as this is the predominant discovery of the manuscript. Translation of mature MyHC isoforms and insertion into the relevant cellular structures is bona fide evidence of maturity. Indeed, the degree of maturity might be quantitatively measured from a ratio of the protein present for immature vs. mature myosin isoforms.

To address this comment, we have performed immunostaining for neonatal MyHC and F-actin and analyzed their subcellular distribution by confocal microscopy. We observed that neonatal MyHC and F-actin showed cross-striation staining pattern, suggesting the presence of neo-MyHC in contractile structures (Figure 2—figure supplement 1A of revised manuscript). We also followed the valuable suggestion of analyzing the protein expression levels of adult *MYHs* by western blot. Whereas *MYH1* and *MYH2* gene expression levels are increased upon combinatorial treatment, we were not able to find detectable levels of adult *MYHs* at the protein level (Figure 2C of the revised manuscript). We have revised the manuscript accordingly to focus on the reliable increased expression of *MYH8*, which is not expressed in early embryonic development but at later stages of development and in the neonatal stage (Cho et al., 1993). Furthermore, we have now validated the protein expression data with antibodies specific for embryonic (MYH3), neonatal (MYH8) and adult (MYH1/2) MHC isoforms, as shown in Figure 2C of the revised manuscript.

In addition, we now include transmission electron microscopy analysis, which show important differences between treated myotubes and respective controls, including the high occurrence of SR-TT junctions and more abundance in mitochondria content.

3) The inhibitors and one activator act on signaling pathways but the authors did not provide evidence for whether the pathways targeted were affected. Moreover, it would be useful to know whether the combinatorial effect was substantially different than individual factors or lesser combinations. These data might provide better knowledge regarding the mechanisms involved as opposed to much further downstream analysis of transcriptomics and epigenetics.

In the initial submission, we have shown thatIngenuity Pathway Analysis (IPA) performed on the RNA-seq data identified the targeted pathways as potential upstream regulators of the differentially expressed genes upon combinatorial treatment (Figure 6—figure supplement 1A). To complement this observation, as suggested by the reviewers, we confirmed by gene expression analysis that the pathways related to each of the small molecules were affected upon combinatorial treatment (Figure 2—figure supplement 2A-D of the revised manuscript).

We have shown that the combinatorial treatment is substantially better than the individual treatment, as evidenced by the quantification of the ratio of area of MyHC/DAPI (Figure 1B and 1C). We agree that a comparison of the effect of each small molecule on the targeted pathways against the combinatorial treatment would reveal whether these pathways are affected differently by the inclusion of additional small molecules. However, we believe that our analysis would then be limited to those four pathways. Instead, we provide whole RNA-seq and ATAC-seq data, allowing us to analyze the data in an unbiased manner. As mentioned above, these analyses revealed that the pathways targeted by the four small molecules were indeed affected in the treated myotubes, as shown in Figure 5—figure supplement 1B (ATAC-seq) and Figure 6—figure supplement 1A (RNA-seq).

4) In the protocol for HTS, the increased ratio of MyHC (+)/DAPI was used as readout to screen for compounds that could improve maturation of PS-derived myotubes. The author's rationale is that statistically significant increase in MyHC/DAPI area ratio in myotubes reflects increased maturation. This approach revealed to be correct, as retrospectively the strategy led to the selection of compounds that actually improve maturation; however, the lack of some details on the experimental protocol does not help on the mechanistic interpretation of the data. For instance, the authors state that the small molecules were added after cells were cultured for three days in the presence of expansion medium (bFGF and Dox), and in concomitance with the switch to differentiation medium, and MyHC expression was evaluated 5 days later. There is no information on whether cells were already differentiated or were proliferating at the time of the beginning of exposure to small molecules. Thus, it remains unclear whether the increased MyHC/DAPI area ratio is due to an effect on cell proliferation, myoblast fusion and/or transcriptional activation of specific genes in post-mitotic myotubes. This also precludes a functional interpretation of the mechanism by which the combined effect of TGFβ, MEK and γ-Secretase inhibition, adenylyl cyclase activation and glucocorticoids exerts the observed effect. As this information seems crucial to further optimize this approach, I encourage the authors to test the effects of these small molecules at sequential stages of PS-derived formation of skeletal muscles, while monitoring cell proliferation, cell fusion and multinucleation with a time course from the beginning of exposure through the whole time of treatment with small molecules.

In line with this valuable suggestion, we analyzed cell proliferation and myotube differentiation through EdU and MyHC staining, respectively, at days 1, 3 and 5, upon switching to terminal differentiation conditions. We observed that proliferation and differentiation in myotubes exposed to DMSO or combinatorial treatment followed a similar pattern at days 1 and day 3. However, we identified dramatic differences at day 5, suggesting that the small molecule treatment predominantly affects the cell proliferation and differentiation between D3 and D5 (Figure 2—figure supplement 1C and 1D of the revised manuscript).

5) Within the same line of point #1, the authors have tested the small molecules in combination (or by subtraction of each of them) always within the same protocol (e.g. exposure at the time of medium switch). It is unknown what is the differentiation status of the cells at that timing, but presumably they are not synchronous, but are rather quite heterogeneous in term of proliferation/differentiation stages. As each of these molecules seems to affect different processes along the sequential stages of skeletal myogenesis, it is highly possible that their optimal synergistic effect can be appreciated by protocols of sequential exposure to each of them, likely with some overlaps of exposures, rather than by an empirical treatment with simultaneous exposure to all compounds, as reported here. The authors should therefore consider evaluating the effect of protocols of sequential exposure in various combinations as compared to simultaneous exposure. As this approach will require multiple experimental points, I would recommend reducing the burden of work by taking one representative endpoint easy to detect – e.g. one representative MYH isoform, as MYH8.

We thank the reviewers for bringing this up. We would like to clarify the steps of the protocol and that cultures of myogenic progenitors are not heterogeneous in terms of differentiation before induction of terminal differentiation. At day 12 of the iPAX7 myogenic differentiation protocol that we describe here, PAX7^+^ myogenic progenitors are sorted from the bulk population (Figure 1—figure supplement 1A). Under PAX7 expression, induced by Dox, the cells maintain a proliferative and non-differentiating state (PAX7^+^). These PAX7^+^ myogenic progenitors are expanded with Dox in the culture media. When myogenic progenitors reach confluency, we switch culture conditions to terminal differentiation, by removing Dox and switching to low nutrient medium, and thus allowing myogenic progenitors to undergo myotube differentiation. In this regard, although we may have ignored the degree of heterogeneity of the myogenic progenitor population at the molecular level, we do know that at day 0 of myotube differentiation (when we also start the combinatorial treatment), the cells are still in a proliferating state and there are no differentiating or differentiated myotubes in the cultures. In fact, Figure 2—figure supplement 1C and 1D of the revised manuscript shows that even at day 1 of myotube differentiation, there are no MyHC (+) myotubes in our cultures, as these begin to arise only at day 3. Therefore, there is no such heterogeneity of proliferating and differentiated cells when we start the small molecule treatment.

As accurately pointed out by the reviewers, we do acknowledge that a potential limitation in the optimization of our combinatorial treatment is that we add all the small molecules at the same time. Following this comment, we have attempted to perform sequential incubations with small molecules (Author response image 1). We started by using a single compound (S, Da, De or F) on D0 and then adding the remaining compounds on D1 or D3. Then, MYH8 protein expression was assessed by western blot. In this first attempt, we did not see a significant difference in any condition when compared to using all the small molecules from D0. Although this attempt was an initial optimization effort that we tried following the suggestion raised by the reviewers, we believe that the number of possible small molecule combinations that can be tried at different sequential and overlapping exposures, and at different days of terminal differentiation, is quite exponentially large, and since we would not be following a particular rational, we would need to try all of these. Thus, we consider this assessment out of the scope of the present study. We do want to emphasize that in spite of the potential for additional optimization, we are confident on the dramatic improvement in differentiation and maturation of differentiated myotubes when performing the combinatorial treatment at day 0, which was validated in many PS cell lines, as shown along the manuscript.

6) More details on the methods used for generating myogenic progenitors in transgene-free conditions should be provided.

We thank the reviewers for this observation. The Materials and methods section has been revised to include the detailed protocol that we have followed for transgene-free myogenic differentiation.

7) The 3D culture data are not sufficiently developed. Are these myotubes more or less mature than the 2D cultures? The Western blot in Figure 6D is relevant for the last two issues raised above as it is clear that neonatal MyHC is a minority of the total MyHC present. This aspect of the work should be further developed or not included as aside from the force measurements, it adds little to demonstrating maturity of the myotubes or aiding in elucidation of the mechanisms involved.

We have further developed the characterization of 3D muscle constructs by immunostaining (Figure 7—figure supplement 1C of the revised manuscript) and by comparing the gene and protein expression of *MYH* isoforms against 2D cultures (Figure 7D and Figure 7—figure supplement 1C of revised manuscript). As observed in Figure 7—figure supplement 1B, 3D cultures showed increased gene expression of all the *MYHs* compared to 2D cultures, except for *MYH7*, which showed a significant decrease. Since MYH7 is expressed even at early stages of muscle development (Cho et al., 1993), the downregulation observed in the presence of S/Da/De/F may not be relevant to maturation. At the protein level, we observed a similar increase in the levels of emb-MyHC and neo-MyHC with combinatorial treatment in both 2D and 3D-cultures when compared to DMSO (Figure 7D). As stated in Materials and methods section, 3D constructs were switched to terminal differentiation conditions for five days. It is possible that a longer timeframe of terminal differentiation would result in a more pronounced increased in maturation, as it has been previously reported (Rao et al., 2018).